biomaterials

multi-substituted hydroxyapatites, Ca, Sr, Mg and P release, release mechanism, Korsmeyer–Peppas model

**Authors for correspondence:**
Gertrud-Alexandra Paltinean
e-mail: pgertrud@gmail.com
Maria Tomoaia-Cotisel
e-mail: mtcotisel.ubbcluj@yahoo.ro

This article has been edited by the Royal Society of Chemistry, including the commissioning, peer review process and editorial aspects up to the point of acceptance.

# Ion release from hydroxyapatite and substituted hydroxyapatites in different immersion liquids: *in vitro* experiments and theoretical modelling study

Aurora Mocanu[1], Oana Cadar[2], Petre T. Frangopol[1],
Ioan Petean[1], Gheorghe Tomoaia[3,4],
Gertrud-Alexandra Paltinean[1], Csaba Pal Racz[1],
Ossi Horovitz[1] and Maria Tomoaia-Cotisel[1,4]

[1]Faculty of Chemistry and Chemical Engineering, Physical Chemistry Centre,
Chemical Engineering Department, Babes-Bolyai University of Cluj-Napoca, 11 Arany J. Street,
400028 Cluj-Napoca, Romania
[2]INCDO INOE 2000, Research Institute for Analytical Instrumentation, 67 Donath Street,
400293 Cluj-Napoca, Romania
[3]Department of Orthopedics and Traumatology, Iuliu Hatieganu University of Medicine and
Pharmacy, 400132 Cluj-Napoca, Romania
[4]Academy of Romanian Scientists, 54 Splaiul Independentei, 050094 Bucharest, Romania

G-AP, 0000-0002-4000-1978; OH, 0000-0002-3652-8558;
MT-C, 0000-0002-0995-3006

Multi-substituted hydroxyapatites (ms-HAPs) are currently gaining more consideration owing to their multifunctional properties and biomimetic structure, owning thus an enhanced biological potential in orthopaedic and dental applications. In this study, nano-hydroxyapatite (HAP) substituted with multiple cations ($Sr^{2+}$, $Mg^{2+}$ and $Zn^{2+}$) for $Ca^{2+}$ and anion ($SiO_4^{4-}$) for $PO_4^{3-}$ and $OH^-$, specifically HAPc-5%Sr and HAPc-10%Sr (where HAPc is HAP-1.5%Mg–0.2% Zn–0.2%Si), both lyophilized non-calcined and lyophilized calcined, were evaluated for their *in vitro* ions release. These nanomaterials were characterized by scanning electron microscopy, field emission-scanning electron microscopy and energy-dispersive X-ray, as well as by atomic force microscope images and by surface specific areas and porosity. Further, the

release of cations and of phosphate anions were assessed from nano-HAP and ms-HAPs, both in water and in simulated body fluid, in static and simulated dynamic conditions, using inductively coupled plasma optical emission spectrometry. The release profiles were analysed and the influence of experimental conditions was determined for each of the six nanomaterials and for various periods of time. The pH of the samples soaked in the immersion liquids was also measured. The ion release mechanism was theoretically investigated using the Korsmeyer–Peppas model. The results indicated a mechanism principally based on diffusion and dissolution, with possible contribution of ion exchange. The surface of ms-HAP nanoparticles is more susceptible to dissolution into immersion liquids owing to the lattice strain provoked by simultaneous multi-substitution in HAP structure. According to the findings, it is rational to suggest that both materials HAPc-5%Sr and HAPc-10%Sr are bioactive and can be potential candidates in bone tissue regeneration.

## 1. Introduction

Calcium phosphate ceramics, such as hydroxyapatite (HAP) and multi-substituted hydroxyapatites, (ms-HAPs), are largely considered for orthopaedic implants and coatings, on metallic implants owing to their excellent biocompatibility, osteoconductive behaviour and chemical stability in the physiological environment [1–8]. Substituted nanocrystalline HAPs with biologically active cations, such as $Mg^{2+}$, $Zn^{2+}$ and $Sr^{2+}$ as substituents for $Ca^{2+}$, and $SiO_4^{4-}$ partially replacing $PO_4^{3-}$ and $OH^-$, in HAP, $Ca_{10}(PO_4)_6(OH)_2$, are considered as valuable sources for a continuous and sustainable release of these important ions [9–23]. The ion release can result from the partial dissolution on the surface of the apatite nanoparticles (NPs) or by an ion exchange with calcium ions from the body fluids. These processes take place through the hydrated layer of loosely bound ions existent on the nanocrystal surface [9].

The role of substituting elements, like Mg, Zn, Sr and Si, in bone regeneration is very well documented [24,25]. These elements could influence bone metabolism, regulate osteoblast and osteoclast activity, stimulate new bone formation, promote osteogenic differentiation of human mesenchymal stem cells and regulate cell activities, such as cell adhesion, migration and proliferation. Therefore, ms-HAP nanomaterials can be considered as multifunctional systems for the release of biologically active ions.

The correlation between ion release from substituted HAP nanomaterials and their effects on cellular activities, as well as their potential toxicity was investigated [26]. Further, the *in vitro* high performance of scaffolds made of substituted HAP materials and their beneficial effects on cellular activities were already confirmed [13,14,21]. Some substituent cations, such as $Ag^+$ [27–29], but $Zn^{2+}$ or $Sr^{2+}$ too, can also induce antibacterial properties, in order to counter possible infections related to the implants [30]. For instance, Sr-doped HAP presented anti-inflammatory effects against macrophages [31].

The present study investigates the effect of adding various amounts of ions to HAP lattice on the ion release properties of the resulting ms-HAP materials into different fluids and on certain physico-chemical characteristics of these ms-HAPs.

Pure stoichiometric HAP, $Ca_{10}(PO_4)_6(OH)_2$, has a very low solubility in water. Its solubility product, noted $K_{sp}$, is an ion activity product given by the following equation:

$$K_{sp} = [Ca^{2+}]^{10}[PO_4^{3-}]^6[OH^-]^2,$$

where the brackets mean the activities of ions. The $K_{sp}$ values given in the literature differ substantially among themselves and strongly depend on the preparation method. However, values of the order of $10^{-117}$–$10^{-118}$, i.e. $pK_{sp}$ values between 117 and 118, are more frequently found, for example, at 25°C, $pK_{sp} = 117$ [32], 117.3 [33] or 116.8 [34]. The theoretical solubility at 37°C would be $8.68 \times 10^{-8}$ mol $l^{-1}$ or 43.56 µg $l^{-1}$ [1].

Actually the 'dissolution' of HAP is a complex process, involving chemical reactions with the formation of different calcium phosphate phases as surface coats [35–37], so no equilibrium state can be reached.

By substitution of some $Ca^{2+}$ ions by other cations, and/or of phosphate and hydroxide ions by silicate or carbonate ions, (but not by fluoride ions), the solubility of substituted HAP is generally increased [10,34,38,39]. For instance, an increase in the solubility of Sr-substituted HAP with increasing strontium content was observed [40]. This aspect could be explained by the perturbation of the HAP lattice, owing to lattice strain provoked by the introduction of foreign ions of different sizes, thus affecting their stability, as well as by the modified crystallinity and crystallite size [41]. Moreover, some of these ions may be at least partially simply adsorbed on the surface of HAP NPs and are

easier displaced in a liquid immersion medium. However, for Mg-substituted HAP, a slowdown of the $Ca^{2+}$ release was also mentioned [42].

For the dissolution process of HAP, a two stage model was proposed [43], involving (i) the surface process, namely the transfer of ions from the crystal surface to the adjacent solution layer, and (ii) the bulk diffusion process, specifically the transfer of ions from this layer into the bulk of the surrounding solution. The rate-controlling step for the dissolution of HAP microcrystals in water at neutral pH was considered as the first stage during the process [43,44].

A dissolution termination could be also explained in terms of a dissolution model that incorporates particle size considerations. It has been suggested and confirmed by experiment at nanoscale level that the dissolution rates decrease markedly with time [33,45], and the dissolution curves might reach a plateau prior to complete dissolution [46].

This theoretical background gives a general framework to support our investigation and provides different models further developed in this study to explain the ion release mechanism from HAP and ms-HAPs into different immersion media as well as to understand the impact of simultaneous cations and anions multi-substitution within HAP lattice on the ion release.

In order to optimize the effects of bone implants which contain ms-HAPs, the availability of their biologically active ions released into the immersion medium should be assessed. To eliminate the influence of complicating biological factors such as the presence of organic components, the ion release study was made by using water and simulated body fluid (SBF).

Our previous results demonstrated that HAP–0.6% Mg–0.2%Zn–0.2%Si [13], HAP-0.47% Si [14] and HAP-Sr, particularly HAP-5%Sr and HAP-10%Sr [21], scaffolds revealed an excellent biocompatibility and high performance in osteoblast cell culture, promoting osteoblast adhesion, growth and proliferation better than unsubstituted HAP. Recently, a starting tailored composition was chosen as a new complex HAP, noted HAPc, namely HAP–1.5%Mg–0.2%Zn–0.2%Si, where the doping elements are in total under 2%, while retaining the HAP structure. Using a composite containing HAPc as coating on Ti implants enhanced the *in vivo* bone consolidation and bone healing [6]. Chemical doping and topographical features as well as nanostructure, appropriate crystallinity and large specific surface area of these nanomaterials are responsible for their *in vitro* and *in vivo* response.

For the aim of the present work, a rational strategy was further used based on specific chemical composition and structure–properties relationship to design new multifunctional HAPs, ms-HAPs, starting with HAPc and adding Sr, mainly HAPc-5%Sr and HAPc-10%Sr, owing to Sr's excellent role on osteoblasts in bone regeneration [24,25]. This working choice is based on the idea that the four-substituted HAP ceramics are expected to inherit the demonstrated beneficial effects owing to HAP structure and to essential elements present in its structure.

Thus, three different cations, $Mg^{2+}$, $Zn^{2+}$ and $Sr^{2+}$, as well as an anion, $SiO_4^{4-}$, were incorporated by partial substitution, into the HAP structure by the wet precipitation route. This substitution confirmed that the resulting nanomaterials, HAPc-5%Sr and HAPc-10%Sr, are very stable, preserving the HAP structure as analysed by X-ray diffraction (XRD), Fourier transform infrared (FTIR) and Raman spectra. Also, some preliminary results on the cation release were briefly illustrated [23].

This study evaluates the *in vitro* release of cations and anions from different nanosized HAP and four-substituted HAP materials, namely HAP-1.5%Mg–0.2%Zn–0.2%Si–5%Sr (HAPc-5%Sr) and HAP-1.5%Mg–0.2%Zn–0.2%Si–10%Sr (HAPc-10%Sr), in both static and simulated dynamic conditions, in water or in SBF. Moreover, the ion release mechanism is also explored. The samples are also characterized by various methods, like scanning electron microscopy (SEM), field emission-SEM (FE-SEM) and energy-dispersive X-ray (EDX), as well as atomic force microscope (AFM), and Brunauer, Emmett and Teller (BET) analysis.

# 2. Experimental procedure

## 2.1. Material and methods

### 2.1.1. Synthesis of hydroxyapatite and multi-substituted hydroxyapatite powders

The investigated HAP and substituted complex HAPs (HAPc-5%Sr and HAPc-10%Sr) were prepared as shown elsewhere [18,23,47,48]. Briefly, calcium nitrate, $Ca(NO_3)_2$, and diammonium hydrogen phosphate, $(NH_4)_2HPO_4$, aqueous solutions in stoichiometric mole ratio Ca/P = 5/3, were the precursors for HAP, $Ca_{10}(PO_4)_6(OH)_2$, preparation, while ammonia solution was used to bring the pH to 11.5. For the substituted HAPs, $Mg(NO_3)_2$, $Zn(NO_3)_2$ and $Sr(NO_3)_2$ were added to the $Ca(NO_3)_2$

solution and tetraethyl orthosilicate (TEOS), $Si(OC_2H_5)_4$, to the phosphate solution, in calculated amounts in order to assure the desired compositions of the end products: HAP containing 1.5 wt% Mg, 0.2 wt% Zn, 0.2 wt% Si and 5%Sr, noted as HAPc-5%Sr, and, respectively, with 1.5 wt% Mg, 0.2 wt% Zn, 0.2 wt% Si and 10% Sr (HAPc-10%Sr).

### 2.1.2. Synthesis of hydroxyapatite

To prepare HAP with the stoichiometric mole ratio of Ca to P, Ca/P = 5/3, equal volumes (5 l) of aqueous solutions, namely 0.25 M calcium nitrate: $Ca(NO_3)_2 \cdot 4H_2O$ (Sigma-Aldrich, Germany) and 0.15 M diammonium hydrogen phosphate: $(NH_4)_2HPO_4$ (Chempur, Poland), both with pH 11.5 adjusted by adding 25% ammonia solution (Chempur, Poland), were mixed at room temperature using a peristaltic pump, in an impact reactor type 'Y'.

### 2.1.3. Synthesis of HAPc-5%Sr

To prepare HAPc-5%Sr, where HAPc is HAP-1.5wt%Mg–0.2wt%Zn–0.2wt%Si, the first solution was obtained by adding 315 ml 0.25 M $Mg(NO_3)_2 \cdot 6H_2O$ solution, 16 ml 0.25 M $Zn(NO_3)_2 \cdot 6H_2O$ solution and 292 ml 0.25 M $Sr(NO_3)_2$ solution (all from Sigma-Aldrich, Germany) to 4377 ml 0.25 M $Ca(NO_3)_2 \cdot 4H_2O$ solution, having the pH adjusted to 11.5 by adding 25% ammonia solution.

The second solution containing the anions (0.15 M in $PO_4^{3-} + SiO_4^{4-}$) was obtained by adding 2.1 ml TEOS (98%, Alfa Aesar, Germany, density 0.933 g $ml^{-1}$) as source for silicate ions, to 4940 ml 0.15 M $(NH_4)_2HPO_4$ solution and water to complete 5 l solution, with adjusted pH 11.5 by adding 25% ammonia solution.

### 2.1.4. Synthesis of HAPc-10%Sr

To prepare HAPc-10%Sr as HAP-1.5wt%Mg–0.2wt%Zn–10wt%Sr–0.2wt%Si, the first solution was obtained by adding 324 ml 0.25 M $Mg(NO_3)_2 \cdot 6H_2O$, 16 ml 0.25 M $Zn(NO_3)_2 \cdot 6H_2O$ and 600 ml 0.25 M $Sr(NO_3)_2$ solutions to 4060 ml 0.25 M $Ca(NO_3)_2 \cdot 4H_2O$ solution.

The second solution containing the anions is the same as that used in the preparation of HAPc-5%Sr. Both solutions had pH 11.5, adjusted by adding 25% ammonia solution

After mixing at room temperature, the obtained suspensions were matured at 22°C (24 h) and then at 70°C (24 h) and the end products, the various HAPs were separated by filtration. Then, precipitates were freeze dried obtaining the lyophilized (non-calcined, nc) powdered samples. Portions of these materials were calcined at 300°C (1 h), to obtain the calcined (calc) samples.

## 2.2. Theoretical formulae for multi-substituted hydroxyapatite nanomaterials

The general theoretical formula for the ms-HAP, is written as $Ca_{10-x-y-z}Mg_xZn_ySr_z(PO_4)_{6-t}(SiO_4)_t(OH)_{2-t}$.

The desired amounts for Mg, Zn, Sr and Si are expressed as functions of $x$, $y$, $z$ and $t$.

$M_{ms-HAP}$ is the molar mass of the multi-substituted HAP:

$$\begin{aligned}
M_{ms\text{-}HAP} &= (10 - x - y - z)A_{Ca} + xA_{Mg} + yA_{Zn} + zA_{Sr} + (6 - t)(A_P + 4A_O) \\
&\quad + t(A_{Si} + 4A_O) + (2 - t)(A_O + A_H) \\
&= M_{HAP} - (x + y + z)A_{Ca} + xA_{Mg} + yA_{Zn} + zA_{Sr} - t(A_P + 4A_O) \\
&\quad + t(A_{Si} + 4A_O) - t(A_O + A_H),
\end{aligned}$$

where $A$ values are the atomic masses (in Daltons), and

$$M_{HAP} = 10A_{Ca} + 6(A_P + 4A_O) + 2(A_O + A_H),$$

is the molar mass of the non-substituted HAP ($x = y = z = t = 0$).

The desired amounts for Mg, Zn, Sr and Si (given in mass per cents, $p$) are expressed as functions of $x$, $y$, $z$ and $t$:

$$p_{Mg} = 100 \, \frac{xA_{Mg}}{M_{ms-HAP}}, \tag{2.1a}$$

$$P_{Zn} = 100 \, \frac{yA_{Zn}}{M_{ms-HAP}}, \tag{2.1b}$$

$$p_{Sr} = 100\,\frac{zA_{Sr}}{M_{ms-HAP}}, \tag{2.1c}$$

$$p_{Si} = 100\,\frac{tA_{Si}}{M_{ms-HAP}}. \tag{2.1d}$$

The system of four equations ((2.1a)–(2.1d)) was solved to find $x$, $y$, $z$ and $t$ for the two substituted HAPs. The general solutions can be written as:

$$x = \frac{M_{HAP}}{A_{Ca} - A_{Mg} + 100(A_{Mg}/p_{Mg}) \times [1 - (A_{Zn} - A_{Ca})p_{Zn}/100A_{Zn} - (A_{Sr} - A_{Ca})p_{Sr}/100A_{Sr} + (A_P - A_{Si} - A_O - A_H)p_{Si}/100A_{Si}]}, \tag{2.2a}$$

$$y = \frac{M_{HAP}}{A_{Ca} - A_{Zn} + 100(A_{Zn}/p_{Zn}) \times [1 + (A_{Ca} - A_{Mg})p_{Mg}/100A_{Mg} - (A_{Sr} - A_{Ca})p_{Sr}/100A_{Sr} + (A_P - A_{Si} - A_O - A_H)p_{Si}/100A_{Si}]}, \tag{2.2b}$$

$$z = \frac{M_{HAP}}{A_{Ca} - A_{Sr} + 100(A_{Sr}/p_{Sr}) \times [1 + (A_{Ca} - A_{Mg})p_{Mg}/100A_{Mg} - (A_{Zn} - A_{Ca})p_{Zn}/100A_{Zn} + (A_P - A_{Si} - A_O - A_H)p_{Si}/100A_{Si}]}, \tag{2.2c}$$

and

$$t = \frac{M_{HAP}}{A_P - A_{Si} - A_O - A_H + 100(A_{Si}/p_{Si}) \times [1 + (A_{Ca} - A_{Mg})p_{Mg}/100A_{Mg} - (A_{Zn} - A_{Ca})p_{Zn}/100A_{Zn} - (A_{Sr} - A_{Ca})p_{Sr}/100A_{Sr}]}. \tag{2.2d}$$

The theoretical formulae resulted from equations ((2.2a)–(2.2d)) for obtained ms-HAP materials are the following:

$$HAPc - 5\%Sr : Ca_{8.76}Mg_{0.63}Zn_{0.03}Sr_{0.58}(PO_4)_{5.93}(SiO_4)_{0.07}(OH)_{1.93}$$
$$HAPc - 10\%Sr : Ca_{8.12}Mg_{0.65}Zn_{0.03}Sr_{1.20}(PO_4)_{5.93}(SiO_4)_{0.07}(OH)_{1.93}.$$

## 2.3. Physico-chemical characterization of hydroxyapatite and multi-substituted hydroxyapatite materials

SEM samples were prepared by deposition of thin films of each specimen by the adsorption from its aqueous dispersion on SEM grids, and finally, a gold layer was sputtered in the Agar Auto Sputter Coater. They were examined on JEOL JSM 5510 LV (Japan), using the secondary electron imaging technique (SE), with accelerating voltage of 30 kV.

FE-SEM, Hitachi SU-8230, operated at 30 kV, was used to explore the nanostructure of HAP samples. FE-SEM is equipped with Oxford energy-dispersive X-ray spectrometer (EDS) for elemental analysis (energy-dispersive X-ray (EDX) spectra). SEM grids were of Cu, covered by a carbon layer of 10–20 nm thickness. SEM samples were prepared by deposition of HAP samples, as powder, in thin layers on SEM grids. FE-SEM coupled to the EDS equipment was also used on ms-HAPs for their elemental analysis by EDX spectra.

An *atomic force microscope* (AFM) JEOL 4210 was used in tapping mode to obtain AFM images for each of the six specimens adsorbed for 10 s from the aqueous dispersions on optically polished glass support [49–51].

Sorptomatic 1990 instrument was used to obtain adsorption–desorption isotherms of nitrogen on the HAP samples dried at 120°C and outgassed at 70°C (6 h), for BET [23].

## 2.4. Ion release from hydroxyapatite and multi-substituted hydroxyapatite materials

For the investigation of ion release, ultrapure deionized water (pH 5.6) [52] and SBF [53] were used as immersion liquids for the HAP materials. The composition of the *Kokubo's SBF* solution was (mmol dm$^{-3}$): Na$^+$ (142.0); K$^+$ (5.0); Mg$^{2+}$ (1.5); Ca$^{2+}$ (2.5); Cl$^-$ (147.8); HCO$_3^-$ (4.2); HPO$_4^{2-}$ (1.0); SO$_4^{2-}$ (0.5), buffered at the physiologic pH 7.40 at 37°C, with tris(hydroxymethyl)amino methane and hydrochloric acid [53]. The ion release was studied in static and simulated dynamic conditions. In both cases, specimens of 0.15 g each material were immersed in 15 ml liquid (10 g l$^{-1}$ HAPs content in the mixture) at 37°C, for various periods of time.

The study in *static conditions* was extended over 90 days. The closed flasks with the HAPs–water and HAPs–SBF mixtures were incubated at 37°C, for 1, 3, 7, 14; 21; 30; 60, and 90 days. After the programmed time, the flask was opened, and the supernatant was filtered (after a prior centrifugation), using 0.45 µm paper filters, and the filtrated solution was analysed for cations and anions content. Three to five samples were tested for HAP and for each ms-HAP material, either non-calcined or calcined powders.

In order to simulate *dynamic conditions*, the immersion liquid in each flask was changed every 24 h, with the same volume of fresh liquid, after separating the supernatant. The test lasted 7 days. This regular replacement approach can be considered a suitable method to simulate *in vitro* the dynamic conditions of a continuous flow of liquid present for a system *in vivo*, avoiding in the same time, the uncertainty related to the liquid rate in the system. At least five samples were tested for HAP and for each ms-HAP material, either non-calcined or calcined powders.

The Ca, Mg, Sr, Zn, P and Si content in the filtrate (immersion liquid) was determined using inductively coupled plasma optical emission spectrometer (ICP-OES) OPTIMA 5300DV (Perkin-Elmer, USA), according to ISO 11885:2009 [54]. The calibration solutions were prepared from multi-element IV storage solutions (for Ca, Mg, Zn, Sr, P and Si). The limits of quantification of the method for the determined cations were 0.008 mg l$^{-1}$ for Ca, 0.005 mg l$^{-1}$ for Mg, 0.005 mg l$^{-1}$ for Zn, 0.010 mg l$^{-1}$ for Sr, 0.05 mg l$^{-1}$ for P and 0.005 mg l$^{-1}$ for Si. These values are lower or at least equal to those requested in ISO 11885:2009.

For dynamic conditions, in the filtrate, on each day of the 7 days, the pH was measured using a multi-parameter analyser pH-meter/conductometer, Multi 350i WTW with SenTix 41 pH electrode, calibrated with buffer solutions (e.g. pH 4 and pH 7) according to SR ISO 10523:2012 [55]. Therefore, at the chosen time, the pH value was recorded in a particular filtrated solution. All data were obtained from five separate experiments for each investigated system, namely water/nanomaterial or SBF/nanomaterial, for all six nanomaterials.

## 2.5. Ion release mechanism

For an understanding of the *release kinetics* of the investigated ions from HAPs, we applied the semi-empirical Korsmeyer–Peppas model [56–61], based on diffusion mechanism. Here, the simple equation (2.3) is used:

$$\frac{M_t}{M_\infty} = kt^n, \tag{2.3}$$

where $M_t/M_\infty$ represents the fraction of the total amount of a species (molecule, element or a specific ion) released at time $t$ from a solid matrix (polymer or crystal lattice). $M_t$ denotes the amount of the species released at time $t$, while $M_\infty$ is the total amount of the chosen species initially present in the solid matrix.

Because the dimensionless $M_t/M_\infty$ ratio appears in equation (2.3), any units for the species amounts can be used, as long as the same units are used for $M_t$ and $M_\infty$. The parameter $k$ of equation (2.3) is a kinetic constant, equal to the released fraction at $t = 1$, so it depends on the units used for time; it is a measure of the release rate. The parameter $n$ is the release constant (diffusional exponent), which characterizes the release mechanism [23]. In the case of Fickian diffusion (diffusion according to Fick's second law), the constant $n$ should present values between 0.43 and 0.5, depending on the shape of the solid matrix (0.43 for spherical particles, 0.5 for thin films [59]). Values from 0.5 to 1.0 are assigned to an anomalous (non-Fickian) transport, while $n = 1.0$ describes a zero-order (time-independent) release rate. Values under 0.43 are also possible, for instance, for poly-dispersed systems [62]. The $k$ and $n$ parameters are obtained from the linearized form of equation (2.3):

$$\log\left(\frac{M_t}{M_\infty}\right) = \log k + n \log t, \tag{2.4}$$

where $n$ is the slope of the straight line of $\log(M_t/M_\infty)$ versus ($\log t$), and $k$ is calculated from the $y$ intercept of the regression line.

## 2.6. Statistical analysis

All measurements were executed in three to five independent experiments, unless otherwise stated, and results are expressed as mean ± s.d.

# 3. Results and discussion

## 3.1. Thin films morphology of hydroxyapatite and multi-substituted hydroxyapatite materials

Morphologic observation revealed the shape and the size of NPs on the surface of the thin films deposited by adsorption from their aqueous dispersions on SEM grids, previously covered with a carbon layer. Film was covered by a thin layer of gold to increase the resolution of SEM images.

The SEM images for the three HAPs, namely HAP, HAPc-5%Sr and HAPc-10%Sr, both non-calcined and calcined ones, are presented in figure 1a–f, along with histograms for the size distribution of their NPs, obtained by measuring the diameters of a great number of particles (greater than 100 particles) in various SEM images.

Calcined HAPc-5%Sr: FE-SEM multi-colour distribution map of elements (figure 1g), EDX spectrum (figure 1h), distribution maps for each element (figure 1i). Gold appears in the EDX spectrum owing to coating of nanomaterial with a gold layer for high imaging resolution.

The SEM images demonstrated the nanostructure of these thin layers, with a porous nano-morphology. All these materials formed almost continuous layers, with various differences in nanoscale morphology, which is porous owing to the interconnections among NPs.

The non-calcined HAP particles (figure 1a) are approximately spherical and relatively distinct, connected in rods, while for HAPc-Sr, they form assemblies associated together to give two-dimensional aggregates (figure 1b,c). The average diameters of the particles are 14.8 ± 4.0 nm for HAP (figure 1a), 14.1 ± 3.1 nm for HAPc-5%Sr (figure 1b) and 13.9 ± 2.9 nm for HAPc-10%Sr (figure 1c), as seen from their histograms (the insets). The average size of particles is given in table 1, for all calcined and non-calcined samples.

For the calcined samples, the aspect of particles is similar, but the particles are somewhat larger, as seen from the histograms (insets) in figure 1d–f. The average diameters of particles are, respectively, 19.5 ± 5.5 nm for HAP (figure 1d), 18.6 ± 4.6 nm for HAPc-5%Sr (figure 1e) and 18.2 ± 3.7 nm for HAPc-10%Sr (figure 1f).

It is to be noted for the non-calcined samples that NPs show the trend to form monodimensional associates, like rods shown in figure 1a–c, while for calcined ones, the assembled NPs in three-dimensional associates are identified, particularly for HAPc-5%Sr (figure 1e). For HAPc-10%Sr, the elongated shapes, like needles, and irregular fragments are found in figure 1f. Further, figure 1e,f reveals that NPs, nanoneedles or nanorods are almost transparent, suggesting that these NPs are very thin, whereas some black NPs are illustrated in figure 1d for calcined HAP and might result from loose associated HAP particles.

Furthermore, the calcined NPs are rather well defined in SEM images (figure 1d–f), indicating that these calcined powders are well dispersed in water and their dispersions have rather good stability. However, the associated agglomerations of cohered NPs are also observed mainly for NPs of calcined HAPc-5%Sr given in figure 1e.

Moreover, the FE-SEM image (figure 1g) showed the surface morphology and unveiled nanoneedles very well packed in a thick layer of calcined HAPc-5%Sr powder deposited on the grid, with the main purpose to determine its corresponding EDX spectrum (figure 1h) for chemical analysis of the calcined HAPc-5%Sr sample.

The multi-colour FE-SEM image (figure 1g) shows the distribution map of elements, Si, Zn, Sr, Ca, P, Mg, O, existing in the ms-HAP, as an example for HAPc-5%Sr calcined nano-powders. The element mapping was used to visualize the elemental composition on the surface of HAPc-5%Sr NPs, which allowed us to image the distribution of the elements across all NPs on the FE-SEM image. The EDX spectrum (figure 1h) and mono-colour distribution maps (figure 1i), jointly display every specific element to distinguish their homogeneous distribution within the ms-HAP layer deposited on the grid.

Thus, the uniform distribution of all elements is evident both in the multi-colour distribution map (figure 1g) and in the individual maps for each element (figure 1i). Similar results were obtained for each as-synthesized nanomaterial, indicating a homogeneous distribution of all elements in these powders.

Subsequently, the chemical composition determined by FE-SEM and EDX on synthesized materials practically coincides with the determined values by ICP-OES [23], which are given in the theoretical formulae. Finally, the incorporation of the four substituting elements into the HAP structure is confirmed. Accordingly, synthesized nanostructured ms-HAP materials might inherit the *in vivo* effects of substituting functional elements, which will eventually be slowly released into the body, and properties of HAP structure for bone healing and regeneration.

Figure 2 shows some representative AFM images for calcined samples, namely for HAP (figure 2a–b), HAPc-5%Sr (figure 2d,e) and HAPc-10%Sr (figure 2g,h). The size of particles on the AFM images is about 41 nm for HAP (figure 2c), 39 nm for HAPc-5%Sr (figure 2f) and 36 nm for HAPc-10%Sr (figure 2i). The higher size of particles on these AFM images, when compared with the values given by the SEM images is probably owing to particle clustering caused by the adsorption and drying process. Their shape appears as nearly spherical, or deformed to have a spherical shape slightly elongated. Similar results were obtained on non-calcined samples. The average diameters of NPs are given in table 1, for all non-calcined and for calcined samples.

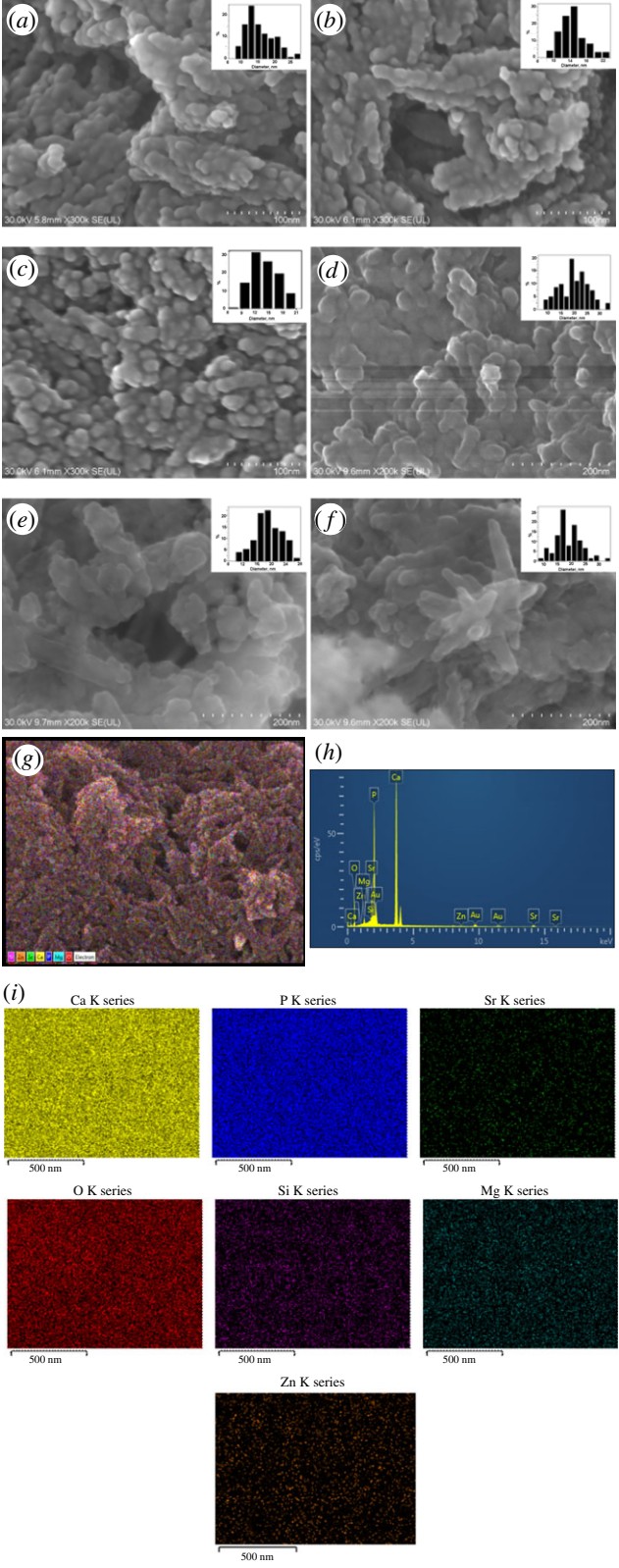

**Figure 1.** SEM images and size distribution of nanoparticles for non-calcined samples: HAP (*a*), HAPc-5%Sr (*b*) and HAPc-10% Sr (*c*), and calcined samples: HAP (*d*), HAPc-5%Sr (*e*) and HAPc-10% Sr (*f*). The bars in the figures are 100 nm (*a–c*) and 200 nm (*d–f*). Calcined HAPc-5%Sr: FE-SEM multi-colour distribution map of elements (*g*), EDX spectrum (*h*), distribution maps for each element (*i*). Gold appears in the EDX spectrum owing to coating of the nanomaterial with a gold layer for high imaging resolution.

**Table 1.** Size of nanoparticles calculated from SSA, and from SEM and AFM images, for all six materials.

| | *d* (nm) | | |
| --- | --- | --- | --- |
| sample | SSA | SEM | AFM |
| non-calcined | | | |
| HAP | 19.8 | 14.8 | 40 |
| HAPc-5%Sr | 17.9 | 14.1 | 38 |
| HAPc-10%Sr | 15.1 | 13.9 | 33 |
| calcined | | | |
| HAP | 22.0 | 19.5 | 41 |
| HAPc-5%Sr | 19.0 | 18.6 | 39 |
| HAPc-10%Sr | 16.7 | 18.2 | 36 |

The roughness of thin films deposited on glass was determined as root mean square (RMS) by AFM on scan area and on cross-section profiles [63]. The RMS values are given in the caption of figure 2. The low values of nano-roughness unveil almost flat thin films of rather well-organized NPs of calcined materials adsorbed on glass. They are rather easily obtained by self-assemblies of constituent NPs and represent a significantly enhanced coating approach. The morphology of ms-HAP self-assemblies turned smoother compared to HAP self-assemblies.

The multi-substitution influenced the size and shape of NPs, their self-assemblies, morphology and porosity. The findings showed that this substitution diminished slightly the size of particles, but did not significantly affect the nanostructure of these materials. The AFM and SEM images demonstrated that the density of self-assemblies made of these NPs within deposited layers on different solids can be controlled by the time elapsed for their adsorption on solids. This result indicates that these films might be used as scaffolds in cell culture and may have potential applications for screening and optimizing HAPs properties for tissue engineering applications.

## 3.2. The specific surface area and porosity of these hydroxyapatite and multi-substituted hydroxyapatite materials

The specific surface area (SSA) and porosity of HAP and ms-HAP powders were evaluated from BET analysis of adsorption–desorption isotherms of these six materials, specifically HAP, HAPc-5%Sr and HAPc-10%Sr, both non-calcined and calcined samples.

As an example, nitrogen adsorption–desorption isotherms and diagrams of pore radius distributions are given in figure 3 for the non-calcined HAPc-5%Sr sample (figure 3*a,b*), and for the calcined HAPc-5% Sr sample (figure 3*c,d*).

The aspect of the isotherms is typical for type IV isotherm in the Brunauer–Demming–Demming and Teller classification, presenting a hysteresis loop. The pore radii are mostly from 2 to 20 nm, thus belonging to the category of mesopores, according to the International Union of Pure and Applied Chemistry notation [64].

These pores should result primarily from the association of NPs as seen in the SEM images (figure 1), while an inner porosity of the NPs is less probable.

The SSAs were calculated as previously shown [23]. The SSA values are rather high, and increase for the non-calcined samples from HAP ($96 \pm 7 \text{ m}^2 \text{ g}^{-1}$) to $106 \pm 5 \text{ m}^2 \text{ g}^{-1}$ for HAPc-5%Sr and $126 \pm 7 \text{ m}^2 \text{ g}^{-1}$ for HAPc-10%Sr. Calcination somewhat reduces these areas, but the order remains the same as for non-calcined samples, namely $86 \pm 6 \text{ m}^2 \text{ g}^{-1}$ for HAP, $100 \pm 7 \text{ m}^2 \text{ g}^{-1}$ for HAPc-5%Sr and $114 \pm 7 \text{ m}^2 \text{ g}^{-1}$ for HAPc-10%Sr. The high SSAs of the nano-HAPs result from the high dispersion of these samples and the inter-particles porosity of the particles associates.

From the SSA, the average size of particles can be roughly estimated, with the assumption that SSA represents the total area of spherical particles with diameter *d*, from 1 g of sample, whose density is $\rho$ [g cm$^{-3}$], by the below equation (3.1):

$$d = \frac{6 \times 10^3}{A\rho} \quad \text{(nm)}, \tag{3.1}$$

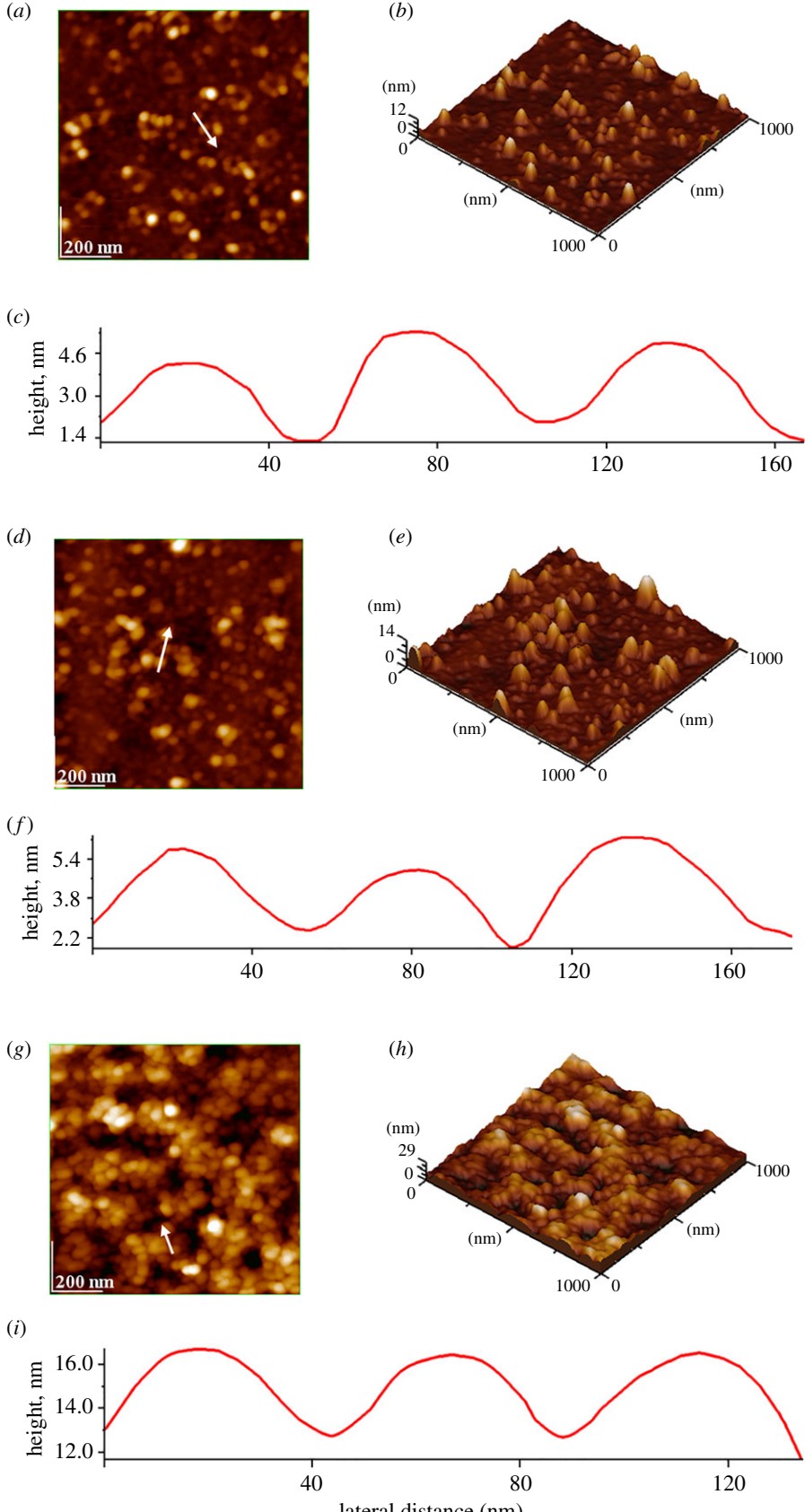

**Figure 2.** AFM images: two-dimensional topographies (*a,d,g*) and three-dimensional images (*b,e,h*) of calcined HAP and ms-HAP samples as thin films, and cross-section profile (*c,f,i*) along the arrow given in two-dimensional topographies for HAP (*a–c*), HAPc-5%Sr (*d–f*) and HAPc-10%Sr (*g–i*); surface roughness (e.g. root mean square: RMS) on area and on profile are, respectively, for HAP, 4.58 nm (*a*) and 1.31 nm (*c*), for HAPc-5%Sr, 1.65 nm (*d*) and 1.29 nm (*f*) and for HAPc-10%Sr, 1.38 nm (*g*) and 1.29 nm (*i*).

(a)

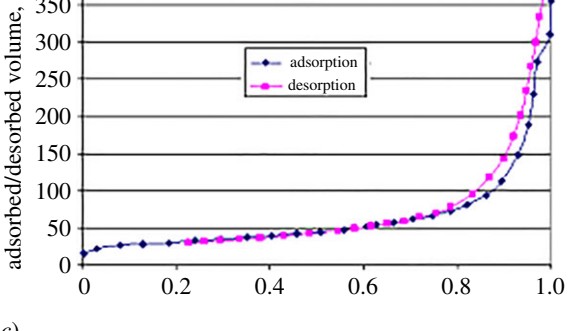

(b)

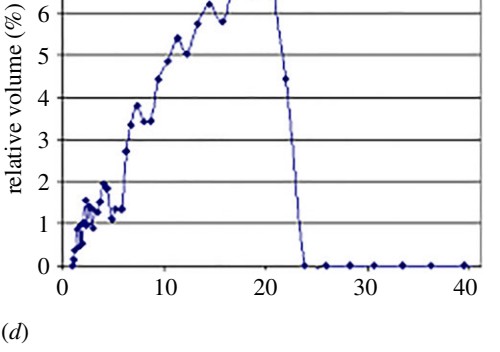

(c)

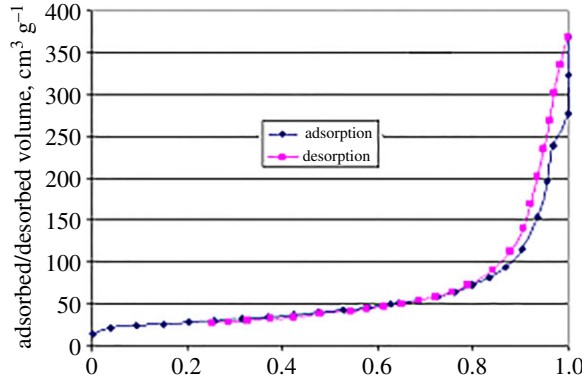

(d)

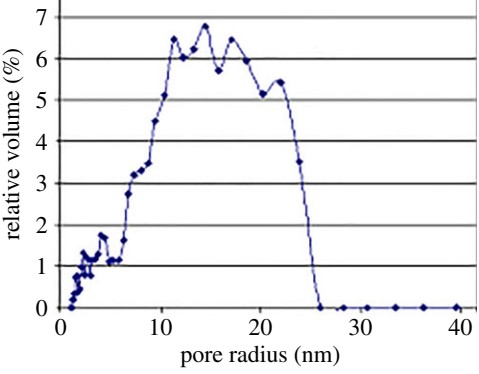

**Figure 3.** Adsorption–desorption isotherms and pore radius distributions for non-calcined HAPc-5%Sr (a,b), and for calcined HAPc-5%Sr (c,d) samples.

where $A$ is SSA in $m^2 g^{-1}$ and $10^3$ is a conversion factor for the units of measurement. The theoretical density of HAP was taken as 3.16 g cm$^{-3}$ [65]. The average diameter, $d$, of the six materials are given in table 1 and compared with the size of particles evaluated from SEM and from AFM images. A similar trend is observed for the size of NPs estimated by the three different experimental approaches.

Further, the multi-element substitution within the HAP lattice might be of great significance for the dissolution behaviour of NPs [45], and can influence the ion release from modified HAPs exposed to various immersion liquids.

## 3.3. Ion release profiles from hydroxyapatite and multi-substituted hydroxyapatites materials, exposed to different liquids

Ion release can be investigated in static conditions, when the immersion medium is not changed for all the duration of the experiment or in dynamic conditions, under a continuous flow of the immersion liquid. While in the first case, the saturation of the liquid (surrounding the solid surface) slows down the release rate, in the second regime, the flow rate affects the release in a manner difficult to control [66]. To avoid this problem in this study, a simulated dynamic regime was applied, consisting of the change of the immersion liquid at each measurement, at given periods of time (i.e. 1 day), with the same quantity of fresh medium.

The release profiles for $Ca^{2+}$, $Mg^{2+}$, $Sr^{2+}$ and for P (phosphate) in water, in static conditions (without changing the medium) for different periods of time up to 90 days are presented in figure 4a, c, e and g, respectively, for HAP and the two complex multi-substituted HAPs, HAPc-5%Sr (noted HAPc-Sr5) and HAPc-10%Sr (named HAPc-Sr10), both non-calcined and calcined samples. The concentrations of ions in the solution are given in mg l$^{-1}$ (ppm). Because the concentrations of $Zn^{2+}$ and silicate ions in the solutions were under the detection limit of ICP-OES, their release could not be examined under these conditions. It is interesting to note that also Sprio et al. [22] did not detect a release of $Zn^{2+}$ ions from substituted HAPs, with a somewhat higher Zn amount than in our materials, in Hank's balanced salt solution. Nevertheless, these ions ($Zn^{2+}$ and silicate) released even as traces cannot be disregarded in

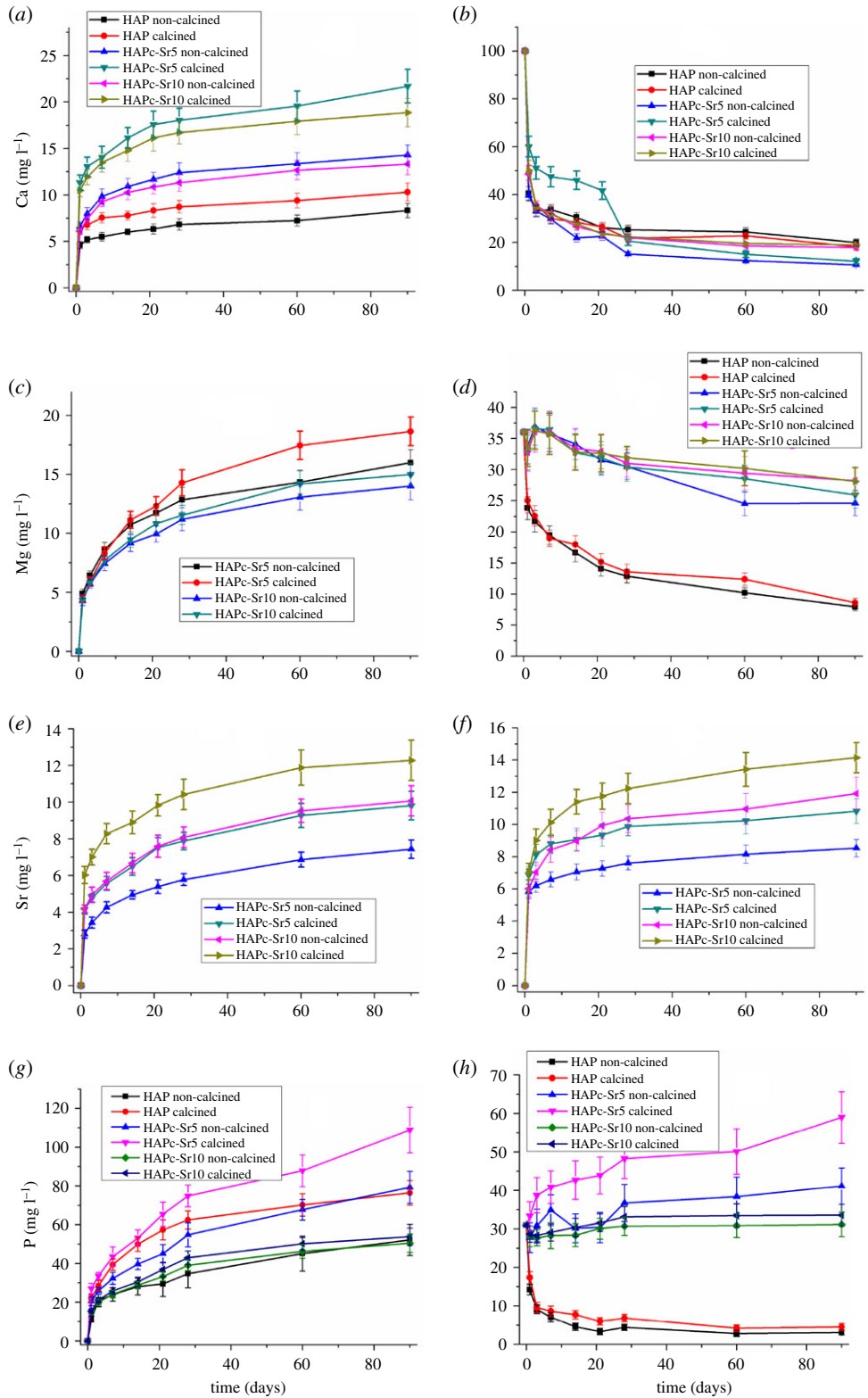

**Figure 4.** Changes of the ions amount in water and in SBF, in the presence of non-calcined and calcined nanomaterials, namely HAP, HAPc-5%Sr and HAPc-10%Sr; $Ca^{2+}$ amount in water (*a*) and in SBF (*b*); $Mg^{2+}$ amount in water (*c*) and in SBF (*d*); $Sr^{2+}$ released amount in water (*e*) and in SBF (*f*); P (phosphate) amount in water (*g*) and in SBF (*h*), in static conditions for 90 days; HAPc denotes HAP-1.5%Mg–0.2%Zn–0.2%Si. Vertical bars represent the standard deviations of measured values.

view of their possible biological effects. As a matter of fact, $Zn^{2+}$ release was observed from Zn-doped HAPs in phosphate buffer solution (PBS), but for much higher Zn content (5–20%) in HAPs [67]. It was also reported that Zn release increased with increasing of Zn amount in the modified HAP samples.

Further in figure 4, the time evolution of $Ca^{2+}$ (figure 4b), $Mg^{2+}$ (figure 4d) and P (figure 4h) concentrations are plotted, after soaking each of the six samples (HAP and ms-HAPs) in SBF, along with $Sr^{2+}$ release in SBF from the Sr-containing complex HAPs (figure 4f).

The cations, $Ca^{2+}$, $Mg^{2+}$ and $Sr^{2+}$ given in figure 4a,c,e, respectively, and P (as phosphate anions) in figure 4g are released in water in a similar way, from each of these samples. The ion amount in solution is continuously increasing in time, but only at the beginning, the released amount is important ('burst release' effect) [22].

For longer time, the released amount diminishes clearly, but no saturation value is attained in these experimental conditions. For all ions and all HAPs, the release from the calcined samples is higher than from the non-calcined ones, but mostly within the limits of experimental errors.

Substituted HAPs, specifically HAPc-5%Sr and HAPc-10%Sr, release higher amounts of calcium ions (figure 4a) and phosphate ions (figure 4g) than pure non-calcined HAP, namely the solubility of HAPs is increased by substitution in the HAP lattice. The higher solubility of substituted HAPs when compared with pure HAP is known in the literature [10,34,38].

The results for Ca release from HAP (figure 4a) are comparable with those for NPs of synthesized anhydrous dicalcium phosphate ($CaHPO_4$) incorporated into a dental resin and soaked in similar static conditions in a NaCl solution for 56 days: $0.35\ mmol\ l^{-1}$ ($14\ mg\ l^{-1}$) [68]. A somewhat higher calcium release from HAP (over $60\ mg\ l^{-1}$) was observed in Hank's balanced salt solution [22].

A particularly high P release is observed for the calcined HAP (figure 4g) in water. The maximum P release is observed for calcined HAPc-5%Sr, followed by the calcined HAP rather close to non-calcined HAPc-5%Sr. Then, a significant decrease in P release is observed for HAPc-10%Sr, which is slightly higher than that for non-calcined HAP. Thus, it seems that the increase in Sr content has a stabilizing effect on the substituted HAP lattice in respect to phosphate ion release.

The release of $Mg^{2+}$ is barely influenced by the higher Sr amount in the complex HAPs (HAPc-5%Sr and HAPc-10%Sr, figure 4c). However, the $Mg^{2+}$ release seems to be slightly higher for calcined HAPc-5% Sr than from HAPc-10%Sr similarly with the phosphate release (figure 4g). As expected, HAPc-10%Sr releases more $Sr^{2+}$ than HAPc-5%Sr (figure 4e).

Some results of size-dependent dissolution kinetics were reported for brushite ($CaHPO_4 \cdot 2H_2O$), which showed that conventional models (e.g. Ostwald–Freundlich model) may be also applied at nanoscale level [69]. The size of particles in our ms-HAPs is slightly decreasing by the progressive substitution in the HAP lattice (table 1). Nevertheless, these size changes are too small to cause a significant increase of solubility, as predicted by conventional models [33]. The notable higher $Sr^{2+}$ release in water (figure 4e) and in SBF (figure 4f) for HAPc-10%Sr compared to HAPc-5%Sr nanomaterials is primarily related to the higher Sr amount in HAPc-10%Sr and not to the smaller size of its NPs.

The dissolution of HAP and of ms-HAPs is an incongruent process [36,37], i.e. the ionic composition released in the liquid phase is different from that in the solid sample immersed (table 2). For HAP, the P/Ca mass ratio in the solution is much above the corresponding ratio in the solid samples. For ms-HAPs, the P/Ca ratio is somewhat smaller, owing to the increased Ca release. The Mg/Ca mass ratio in water in the presence of non-calcined HAPc-5%Sr varies from 0.74 (the first day) to 1.12 (the 90th day), and from 0.39 to 0.86 for the calcined sample, while in the solid, the ratio is 0.044. For HAPc-10%Sr in water, the Mg/Ca mass ratio varies from 0.71 to 1.05 (non-calcined sample), and from 0.42 to 0.65 (calcined sample), against 0.048 in the solid. It is evident that the deviation from the solid phase composition increases in time, and is higher for the non-calcined samples versus the calcined ones.

For Sr release, the differences between the Sr/Ca ratio in the solution (water) and in the solid are less striking, but the concentration of $Sr^{2+}$ in the liquid remains higher, as expected from its content in the solid. The Sr/Ca ratio is about 0.4–0.5 in the solution against 0.15 in the solid for HAPc-5%Sr and around 0.6–0.8 versus 0.32 for HAPc-10%Sr. In this case again, the Sr excess increases in time. The Sr release was also connected with the presence of a large amount of Sr in the amorphous phase of HAPs [70].

The Mg/Sr mass ratio is always much higher in the aqueous solution than in the solid ms-HAPs. This confirms that Mg is released the easiest, as mentioned also in the literature [71,72]. A high release of Mg was likewise observed from Mg-HAP scaffolds in PBS [73]. This intense $Mg^{2+}$ release generally improves the substituted HAPs affinity with cells as well as their bioactivity [13,22,73].

For the samples immersed in SBF, the ion release evolution is more complex (figure 4b,d,f,h), than in the case of ion release in water, for static conditions. Only the strontium release profile (figure 4f) goes likewise as in water (figure 4e), because SBF does not contain $Sr^{2+}$ ions. Consequently, no precipitation of such ions can be expected on the surface of NPs of ms-HAPs calcined or non-calcined on immersed samples in SBF.

**Table 2.** Mass ratio between elements in aqueous phase compared with the ratio in the immersed solid sample, for HAP and ms-HAPs for non-calcined and calcined samples; the ion release was monitored in static and in simulated dynamic conditions. (nc, non-calcined; calc, calcined; cum, cumulated for 7 days.)

| ratio of elements | method | phase | day | HAP nc | HAP calc | HAPc-5% Sr nc | HAPc-5% Sr calc | HAPc-10% Sr nc | HAPc-10% Sr calc |
|---|---|---|---|---|---|---|---|---|---|
| P/Ca | | solid | — | 0.46 | 0.46 | 0.52 | 0.52 | 0.57 | 0.57 |
| | static | water | 1 | 2.46 | 3.45 | 3.11 | 2.38 | 2.50 | 1.47 |
| | | | 90 | 7.46 | 7.42 | 5.54 | 4.65 | 4.54 | 3.38 |
| | dynamic | | 7 | 2.81 | 2.34 | 1.58 | 1.42 | 2.56 | 1.71 |
| | | | cum | 2.60 | 3.08 | 2.54 | 2.33 | 2.34 | 2.10 |
| Mg/Ca | | solid | — | — | — | 0.044 | 0.044 | 0.048 | 0.048 |
| | static | water | 1 | | | 0.74 | 0.39 | 0.71 | 0.42 |
| | | | 90 | | | 1.12 | 0.86 | 1.05 | 0.65 |
| | dynamic | | 7 | | | 0.68 | 0.44 | 0.63 | 0.57 |
| | | | cum | | | 0.59 | 0.56 | 0.64 | 0.58 |
| Sr/Ca | | solid | — | — | — | 0.15 | 0.15 | 0.32 | 0.32 |
| | static | water | 1 | | | 0.42 | 0.37 | 0.69 | 0.58 |
| | | | 90 | | | 0.52 | 0.45 | 0.75 | 0.65 |
| | dynamic | | 7 | | | 0.61 | 0.36 | 0.69 | 0.77 |
| | | | cum | | | 0.46 | 0.38 | 0.64 | 0.73 |
| Mg/Sr | | solid | — | — | — | 0.30 | 0.30 | 0.15 | 0.15 |
| | static | water | 1 | | | 1.73 | 1.07 | 1.03 | 0.72 |
| | | | 90 | | | 2.15 | 1.90 | 1.39 | 1.17 |
| | dynamic | | 7 | | | 1.11 | 1.24 | 0.92 | 0.74 |
| | | | cum | | | 1.27 | 1.49 | 0.99 | 0.80 |

The released amount of strontium is, however, higher in SBF (figure 4f) than in water (figure 4e), as observed also by other authors [74]. This situation might be owing to the contribution of ionic exchange of $Sr^{2+}$ from the lattice of ms-HAPs with $Ca^{2+}$, [9] and perhaps with $Mg^{2+}$, which are present in SBF. Again, the samples richer in Sr (10%Sr) release a higher amount of $Sr^{2+}$. After two to three weeks, the profile of Sr release becomes nearly linear, i.e. the release rate remains almost constant. A similar observation was made [75] for micrometric Sr-substituted HAP in SBF, after one week. A study of ion release from Sr, Zn, Ag and F substituted HAPs in SBF [26], in static conditions, for 7 and 14 days, also reported a higher $Sr^{2+}$ release for samples with 5% Sr when compared with those with 2.5% Sr. For $Zn^{2+}$, a small amount was released from samples with 2.5% Zn, and quite a bit more from samples with 5% Zn, both materials having a much higher Zn amount than our samples. For both, Sr and Zn, the ion release increased in time.

The $Ca^{2+}$ and $Mg^{2+}$ concentrations in SBF display an overall decreasing trend from their initial amount in SBF, namely from 100 mg l$^{-1}$ $Ca^{2+}$ (figure 4b) and from 36 mg l$^{-1}$ $Mg^{2+}$ (figure 4d), denoting an uptake of these ions (into nanomaterials) from SBF solution, through ion adsorption and ion exchange.

For calcium, the decrease is very fast in the first days (figure 4b), but after about 20 days, the concentration stabilizes with a slow decrease, similarly for all six nanomaterials, whereby the $Ca^{2+}$ levels are somewhat lower for substituted HAPs than for pure HAP. A similar $Ca^{2+}$ amount of decrease was observed [76] in SBF in the presence of HAP, as well as Si and carbonate substituted HAPs, from 100 to 50–30 ppm, with a stabilization and even a small increase after three weeks. A higher Ca release from substituted HAPs was expected, but the ion exchange of $Ca^{2+}$ from SBF solution with $Mg^{2+}$ and $Sr^{2+}$ from the solid NPs could counteract this effect.

Reasonably, we can assume that two opposed processes take place when HAP or substituted HAPs are in contact with SBF: (i) the formation of a new HAP resulting from the ions adsorbed ($Ca^{2+}$, $Mg^{2+}$ and phosphate) from SBF, on the surface of a particular nanomaterial solid phase, which is responsible for the ions decreasing concentration in SBF solution; and (ii) the release of ions by dissolution from the chosen

solid phase, which slows down the ion deletion in SBF. The second process should be more intense from substituted HAPs, as seen especially for $Mg^{2+}$ in figure 4*d*, and for phosphate ions in figure 4*h*, so in their presence, more $Mg^{2+}$ and phosphate ions remain in SBF solution. The final result is, however, a net diminution of the concentration of these $Ca^{2+}$, $Mg^{2+}$ and phosphate ions in the SBF solution in the presence of non-substituted HAP.

The release of $Ca^{2+}$ from nano-HAP in a SBF solution, without $Ca^{2+}$ in its composition (containing only $Na^+$, $K^+$ and $Mg^{2+}$), [77] was from about 55 ppm after 2 days to about 63 ppm after 15 days, i.e. higher than the values observed by us (about 30 ppm at 15 days, figure 4*b*) in SBF with $Ca^{2+}$ in its composition of about $100\ mg\ l^{-1}$. Therefore, the $Ca^{2+}$ amount precipitated from SBF in our experiments should largely exceed the values of $Ca^{2+}$ release observed by Fathi *et al.* [77].

The uptake of $Ca^{2+}$ ions from SBF was also observed on HAP, and on silicon and carbonate substituted HAPs discs, and the formation of new HAP from the adsorbed $Ca^{2+}$ and phosphate ions was evidenced [76].

An important decrease of P amount in SBF solution (from its initial value of $31\ mg\ l^{-1}$ to about $5\ mg\ l^{-1}$) is observed in the presence of the non-substituted HAP (figure 4*h*). The high uptake of P from SBF by HAP samples was also observed on related systems [76], even until a complete consumption of P. For the calcined HAPc-5%Sr nanomaterial, there is a significant increase of the P amount in the SBF solution, denoting that the high P release observed from the same nanomaterial in the presence of water (figure 4*g*) exceeds the P deposition as a new HAP on that solid. For the HAPc-10%Sr nanomaterials, either calcined or non-calcined, the P amount is nearly constant (at slightly above $30\ mg\ l^{-1}$). It seems to be an equilibrium between P release from HAPc-10%Sr NPs and P uptake from SBF, at a nearly constant value of pH 7.45 (figure 6*b*), in these experimental conditions.

Further, we suggest that HAPc-10%Sr nanomaterials can be used as scaffolds and might have the capacity to regulate pH to an alkaline level in Dulbecco's Modified Eagle Medium (DMEM), which is a widely used basal medium for cell culture, and thus, favouring cell bioactivity.

The decrease of $Mg^{2+}$ amount in the SBF solution is very strong (figure 4*d*) in the presence of HAP, which initially does not contain any magnesium. The $Mg^{2+}$ ions from SBF are deposited on the HAP solid NPs by precipitation with phosphate ions and/or by ionic exchange with $Ca^{2+}$ ions from the HAP lattice.

In the presence of substituted HAPs, the decrease of the magnesium amount in SBF is limited by the release of $Mg^{2+}$ from the solid NPs phase, and maybe again by $Mg^{2+}$ ion exchange with $Ca^{2+}$ ions and other cations from SBF [9,78]. An increased dissolution of magnesium-substituted fluoro-apatite in SBF with higher Mg content in the nano-powders is reported in the literature, along with a more intense precipitation of newly formed HAP on their surface [79].

The results for $Ca^{2+}$, $Mg^{2+}$ and phosphate release from HAPs in water and for $Sr^{2+}$ in water and SBF, in simulated dynamic conditions, are shown in figure 5 as plots of the daily release (figure 5*a,c,e,g,i*) and of the cumulative release in 7 days (figure 5*b,d,f,h,j*).

Rationally, by the daily change of the immersion liquid (i.e. simulated dynamic conditions), the amount of released ions was much higher (figure 5) than in static conditions (figure 4), in which the ion release was slowed down, at a longer time period, by the accumulation of ions in the surrounding solution. This is evident from the comparison of the plots for cumulated ion release in dynamic conditions (figure 5*b,d,f,h,j*) and the corresponding plots for ion release in static conditions (figure 4*a,c,e,f,g*). The daily release rate for the ions (figure 5*a,c,e,g,i*) is the highest in the first 2 days (in some cases, even higher in the second day than in the first day). It next decreases, but then in static conditions with a trend to almost constant rate in the last days. As well as in static conditions, $Ca^{2+}$ release (figure 5*a,b*) and phosphate release (figure 5*i,j*) are higher from substituted HAPs than for pure HAP, and for calcined samples than for non-calcined. The release of $Mg^{2+}$ from ms-HAPs (figure 5*c,d*) seems to be little influenced by the Sr amount (5% or 10%) within the nanomaterials. However, the $Mg^{2+}$ release profile remained higher (figure 5*d*) for calcined HAPc-5%Sr than for calcined HAPc-10%Sr similarly with the situation found in static release conditions (figure 4*c*). The $Sr^{2+}$ release is more important from HAPc-10%Sr than from HAPc-5%Sr, both in water (figure 5*e,f*) and in SBF (figure 5*g,h*). Like in static conditions, the amount of strontium released in SBF is much greater than in water.

The phosphate cumulative release (figure 5*j*) appears to be slightly greater in simulated dynamic conditions from calcined HAPc-5%Sr than from HAPc-10%Sr similarly with the phosphate release in static conditions (figure 4*g*).

From the examination of the ion release profiles in figure 5*a,c,e,g,i*, a two-step release could be inferred, in substantial agreement with related data [80,81]. Therefore there is a more rapid initial

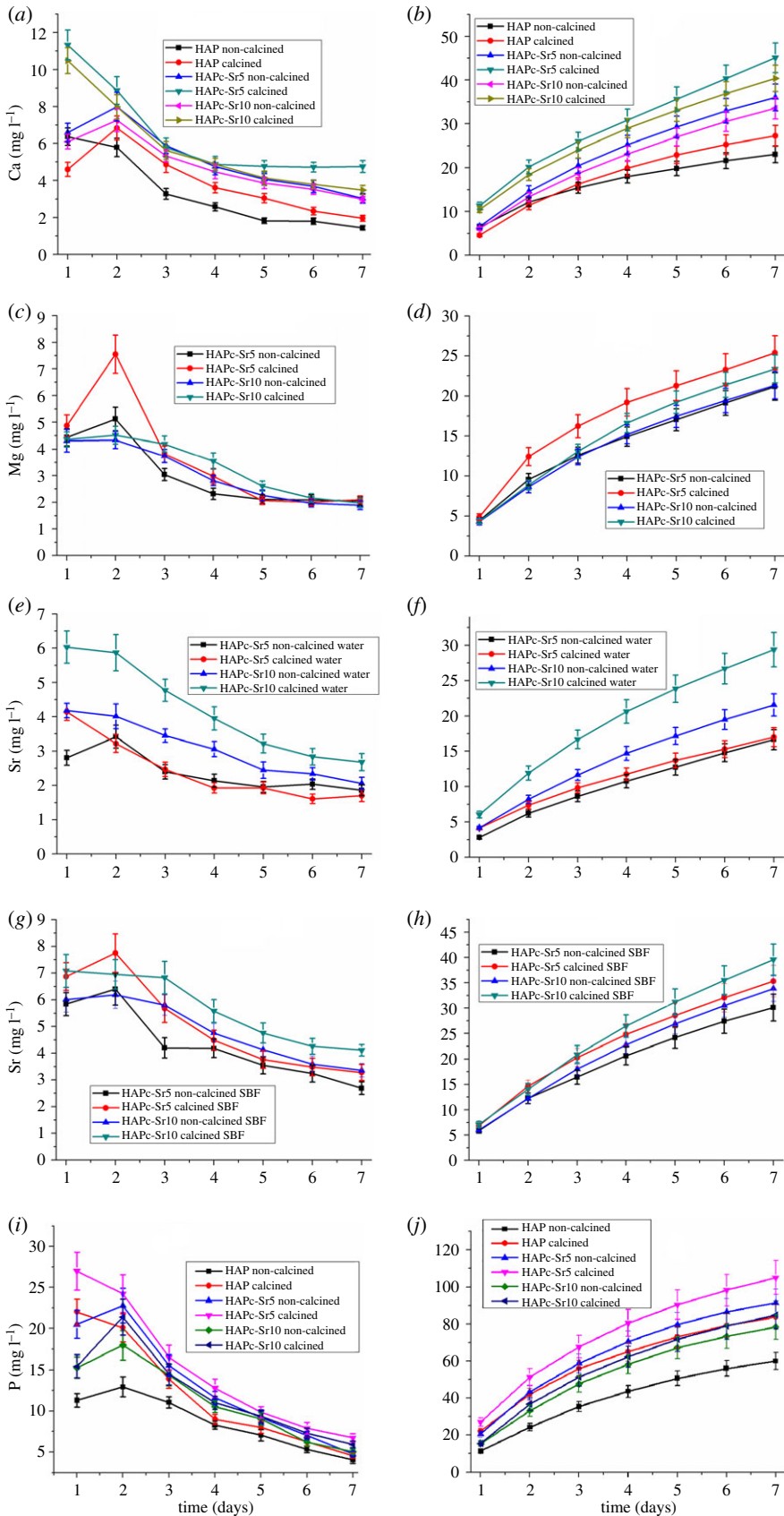

**Figure 5.** Daily variation of Ca (*a*), Mg (*c*), Sr (*e*) and P (*i*) amount in water and Sr (*g*) amount in SBF, in the presence of HAP, HAPc-5%Sr and HAPc-10% Sr; non-calcined and calcined samples; corresponding cumulative release of Ca (*b*), Mg (*d*), Sr (*f*) and P (*j*) in water and Sr in SBF (*h*). Vertical bars represent the standard deviations of measured values. The immersion liquid was changed every day to simulate dynamic conditions.

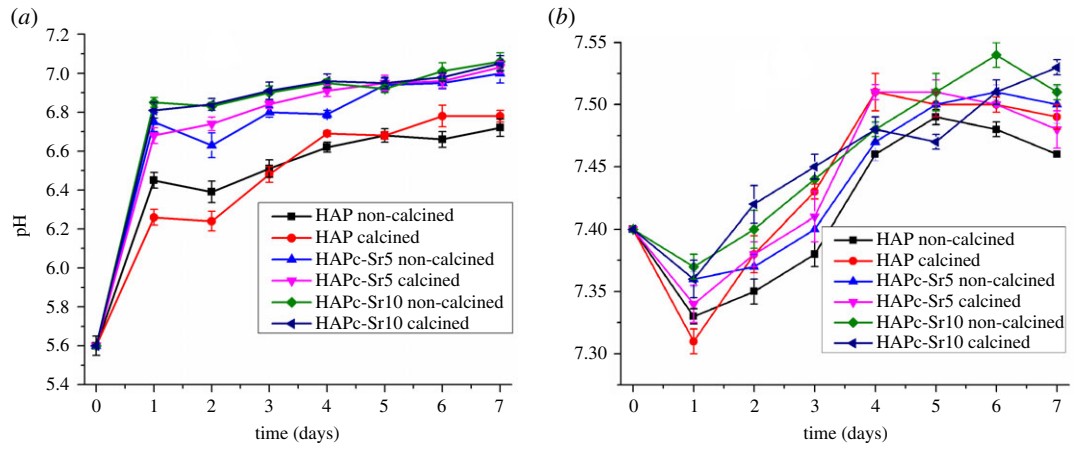

**Figure 6.** The pH variation in water (*a*) and in SBF (*b*) in the presence of HAP and substituted HAPs, i.e. HAPc-5%Sr (noted HAPc-Sr5) and HAPc-10%Sr (HAPc-Sr10), both non-calcined and calcined samples, for various time periods in simulated dynamic conditions; HAPc denotes HAP-1.5%Mg–0.2%Zn–0.2%Si. The vertical bars represent the standard deviations of the measured values.

release ('burst' release) of the ions at the surface of HAP NPs and their diffusion into the surrounding liquid, followed by the slower inner diffusion of these ions in the bulk of NPs to the solid/liquid interface.

The same incongruent dissolution in water is observed in simulated dynamic conditions as in static situations (table 2), both for the daily release and for the total cumulated release, with a favoured $Mg^{2+}$ and $Sr^{2+}$ release.

From the $Sr^2$ ion release of the ms-HAPs, some important remarks can be inferred regarding their physiological effect and potential medical application. The results revealed a steady release of $Sr^{2+}$ ions from HAPc-Sr nanomaterials in water (figure 4*e*) and in SBF (figure 4*f*). At 90 days of immersion in static conditions, the concentration of $Sr^{2+}$ ions released from HAPc-5%Sr and HAPc-10%Sr, both non-calcined and calcined nanomaterials, in water was within the range of 7–12 ppm, as shown in figure 4*e*. The $Sr^{2+}$ ions released from these nanomaterials in SBF was in the concentration range of 9–13 ppm (figure 4*f*). Significantly higher $Sr^{2+}$ amounts were released in simulated dynamic conditions, at 7 days, from HAPc-Sr nanomaterials, as illustrated in figure 5*f* (in water) and in figure 5*h* (in SBF). It is important to emphasize that the Sr release at these concentrations in immersion liquids (static and dynamic regime) is within the therapeutic window of Sr concentrations of 2–45 ppm, [82,83] and subsequently, it is expected to promote osteogenesis and *in vivo*, bone regeneration.

Furthermore, the concentration of the other ions released in water in static conditions from these nanomaterials, at 90 days, was in the following range: $Ca^{2+}$ release: 8–21 ppm (figure 4*a*); $Mg^{2+}$ release: 14–18 ppm (figure 4*c*); P release: 50–109 ppm (figure 4*g*), while higher amounts of these ions were released in water, in simulated dynamic conditions from these nanomaterials, at 7 days (figure 5*b,d,j*).

It appears reasonable to conclude that the ion release profiles can be controlled by adjusting the amount of each substituting element initially introduced in the HAP structure. Also these HAPc-Sr materials highlight the suitability of the multi-substitution strategy within HAP structure, to tailor advanced ms-HAP nanomaterials, obtaining proper different release profiles depending on the requirements for a chosen ion, and accordingly to the ion release kinetics. The pure HAP is a primary material suitable for functionalization by substitution with various elements directing for improvements on physico-chemical and biological properties and finally leading to novel graft materials. The understanding is needed on how such structural changes can lead to improved materials for therapeutic applications

## 3.4. Changes of pH in immersion media exposed to hydroxyapatite and multi-substituted hydroxyapatites

The pH changes in each day of the 7 days are displayed in figure 6*a,b*, for the immersion of the HAP and ms-HAP powders in water and in SBF, respectively, in simulated dynamic conditions.

In water, there are fluctuations in the pH values from day to day (figure 6*a*), but there is an overall tendency of pH increasing in time, i.e. the alkalinity of the surrounding medium is slightly increased after

the interaction of the medium with the dispersed HAP and ms-HAP samples. With substituted HAPs, the pH is something higher than for pure stoichiometric HAP. It appears that the dissolution of ms-HAP (HAPc-5%Sr and HAPc-10%Sr) samples brings about a slight alkalization of the aqueous solutions.

As shown in figure 6*b*, the change of pH in SBF appears similar for all six samples. The pH is slightly reduced in the first day for all samples from pH 7.40 (which corresponds to buffered SBF solution) to the low value of pH 7.31 for calcined HAP and to the high value of pH 7.37 for non-calcined HAPc-10%Sr. After the first day, a minor tendency to increased pH values in time, for subsequent days can be observed. After the fifth and the sixth day, only small fluctuations of the pH values are observed. At the seventh day, pH appears to be stabilized between about 7.46 for non-calcined HAP and 7.53 for calcined HAPc-10%Sr.

A similar trend was observed on green and heat-treated HAP and, Si and carbonate substituted HAP discs in SBF [77], and on HAP and HAP-Mg composite scaffolds in PBS with partial replacement of the PBS after each measurement [73].

Moreover, analogous results were previously described for the changes of pH of cell culture medium, DMEM, in contact with HAP-Cu, and pH remained at 7.2, and inorganic phosphate concentration was around 1.4 mM, which were designated to be compatible with cell viability and proliferation [84].

We suggest that a similar behaviour of HAP, HAPc-5%Sr and HAPc-10%Sr nano-powders, as that discovered in SBF, could be extended to cell culture medium. In consequence, the settled environmental pH and also maintaining the ion released amounts, within a cell friendly range, recommend these HAP, HAPc-5%Sr and HAPc-10%Sr materials as scaffolds and coatings on metallic implants for therapeutical applications.

## 3.5. Ion release kinetics using the Korsmeyer–Peppas model

The Korsmeyer–Peppas equations (2.3) and (2.4) were applied to the experimental data for ion release in static conditions, for 1–90 days (table 3), and in simulated dynamic conditions, for 1–7 days (table 4). The values for the kinetic constant $k$ and the diffusional exponent (named also release constant) $n$ are given with their standard errors from the regression analysis, along with the coefficients of determination, $r^2$. The $M_\infty$ values for Ca were calculated from the formulae of the HAPs samples, for a sample weight of 10 000 mg (10 g), corresponding to 1 l solution, having in view that the released amount of ions is given in mg l$^{-1}$ (ppm), as in figures 4 and 5. The $M_\infty$ values for Ca and P were, respectively, 3990 and 1850 mg for HAP, 3430 and 1800 mg for HAPc-5%Sr, 3100 and 1760 mg for HAPc-10%Sr. For Mg, $M_\infty$ is 150 mg in both complex HAPs (with 1.5% Mg), and for Sr, it is 500 mg in HAPc-5%Sr and 1000 mg in HAPc-10%Sr. For the simulated dynamic conditions, $M_t$ values were the cumulated release values for 1–7 days.

For illustration, some regression lines for ion release from non-calcined HAPc-10%Sr are given in figure 7. The high values of the coefficients of determination, $r^2$, for ion release in static conditions (0.96–0.99) prove that the Korsmeyer–Peppas model is suitable for describing this process (table 3). The values of the kinetic constant, $k$, related to the extrapolated released fraction for the first day are of the order of magnitude of $10^{-3}$–$10^{-2}$. They confirm the observations made from inspecting the release curves given in figure 4 and the data displayed in table 2, namely higher release for $Mg^{2+}$, followed by phosphate and $Sr^{2+}$, and a slower release for $Ca^{2+}$. More strontium is released in SBF than in water, and more calcium from the ms-HAPs than from pure HAP.

The release constant (diffusional exponent), $n$, shows surprisingly low values of about 0.1–0.3, lower than the 0.43 value characteristic for Fickian diffusion [59,62]. This could be an indication that diffusion is not the rate-determining step for ion release, when the solution is not replaced in time. The accumulation of $Ca^{2+}$ and $PO_4^{3-}$ ions at the interface between NPs of HAPs and liquid and the interfacial reactions with the surrounding medium can lead to the formation of an adjacent layer of low permeability which reduces the diffusion of ions from the solid to the solution and diminishes the dissolution rate, leading to a self-inhibiting dissolution [85]. Values of the diffusional exponent, $n$, under 0.5 were interpreted as an indication for a hindered (or retarded) diffusion [86]. Similar low $n$-values (even under 0.1) were also found for the release of vancomycin from calcium phosphates in static conditions [65]. Further, for the release of $Sr^{2+}$ from a Sr-loaded bone cement containing HAP, in α-Minimum Essential Medium Eagle culture media in static conditions for 10 days, a $n$-value of about 0.32 was found [87].

The situation is quite different when the immersion liquid is changed each day (table 4 and figure 8). The linearity of the experimental points according to equation (2.4) is still good ($r^2$ values from 0.96 to 0.99, except a

**Table 3.** Korsmeyer–Peppas model, kinetic constant, $k$, and release constant, $n$, of ions released from HAP and ms-HAP nanomaterials in water and in SBF, in static conditions (for 90 days).

| element | sample | immersion fluid | $10^3 \, k$ (day$^{-n}$) | $n$ | $r^2$ |
|---|---|---|---|---|---|
| Ca | HAP non-calcined | water | $1.122 \pm 0.028$ | $0.124 \pm 0.010$ | 0.9596 |
| | HAP calcined | | $1.542 \pm 0.035$ | $0.105 \pm 0.008$ | 0.9636 |
| | HAPc-5%Sr non-calcined | | $1.968 \pm 0.045$ | $0.174 \pm 0.008$ | 0.9858 |
| | HAPc-5%Sr calcined | | $3.24 \pm 0.06$ | $0.144 \pm 0.006$ | 0.9862 |
| | HAPc-10%Sr non-calcined | | $2.024 \pm 0.048$ | $0.175 \pm 0.008$ | 0.9855 |
| | HAPc-10%Sr calcined | | $3.367 \pm 0.043$ | $0.1346 \pm 0.0043$ | 0.9929 |
| P | HAP non-calcined | water | $6.79 \pm 0.45$ | $0.311 \pm 0.023$ | 0.9636 |
| | HAP calcined | | $11.96 \pm 0.54$ | $0.291 \pm 0.015$ | 0.9813 |
| | HAPc-5%Sr non-calcined | | $10.47 \pm 0.50$ | $0.307 \pm 0.016$ | 0.9808 |
| | HAPc-5%Sr calcined | | $13.80 \pm 0.65$ | $0.315 \pm 0.016$ | 0.9822 |
| | HAPc-10%Sr non-calcined | | $8.38 \pm 0.26$ | $0.274 \pm 0.010$ | 0.9899 |
| | HAPc-10%Sr calcined | | $8.65 \pm 0.30$ | $0.287 \pm 0.012$ | 0.9885 |
| Mg | HAPc-5%Sr non-calcined | water | $33.3 \pm 1.1$ | $0.270 \pm 0.012$ | 0.9861 |
| | HAPc-5%Sr calcined | | $29.0 \pm 1.1$ | $0.338 \pm 0.013$ | 0.9899 |
| | HAPc-10%Sr non-calcined | | $29.1 \pm 0.7$ | $0.270 \pm 0.008$ | 0.9940 |
| | HAPc-10%Sr calcined | | $29.3 \pm 0.7$ | $0.285 \pm 0.008$ | 0.9948 |
| Sr | HAPc-5%Sr non-calcined | water | $5.53 \pm 0.05$ | $0.2207 \pm 0.0033$ | 0.9984 |
| | HAPc-5%Sr calcined | | $7.87 \pm 0.22$ | $0.204 \pm 0.010$ | 0.9845 |
| | HAPc-10%Sr non-calcined | | $4.57 \pm 0.11$ | $0.205 \pm 0.008$ | 0.9892 |
| | HAPc-10%Sr calcined | | $5.96 \pm 0.08$ | $0.164 \pm 0.005$ | 0.9939 |
| Sr | HAPc-5%Sr non-calcined | SBF | $11.38 \pm 0.16$ | $0.0862 \pm 0.0046$ | 0.9801 |
| | HAPc-5%Sr calcined | | $14.29 \pm 0.26$ | $0.094 \pm 0.006$ | 0.9694 |
| | HAPc-10%Sr non-calcined | | $6.08 \pm 0.12$ | $0.152 \pm 0.007$ | 0.9867 |
| | HAPc-10%Sr calcined | | $5.72 \pm 0.22$ | $0.192 \pm 0.013$ | 0.9679 |

0.93 value for Mg release from non-calcined HAPc-5%Sr in water), but the release constant, $n$, is much greater than in static conditions. For the ion release, the $n$-values vary from 0.64 to 0.90.

Such an increase of the release constant, $n$, by passing from static to dynamic conditions was also indicated for drug release from calcium phosphates, [66] but depending on the liquid flow rate. This situation is classified as non-Fickian transport, and could be understood as a combination of diffusion and dissolution of the material, in agreement with similar data in the literature [12]. For strontium release in SBF, a contribution of ionic exchange cannot be excluded.

A thorough inspection of the regression lines for the release in simulated dynamic conditions (figure 8) reveals a systematic deviation of the points for the first day (or for the first 2 days) from the regression line. Considering only the release in the last 4 and 5 days, a better correlation is found ($r^2$ values near to 1) and substantial lower $n$-values (0.5–0.7), nearer to the assumed value for Fickian diffusion.

In the following, some examples are given: for $Ca^{2+}$ from non-calcined HAPc-10%Sr, days 3–7, $n = 0.69$, with $r^2 = 0.9982$; for $Mg^{2+}$ from the same sample, in days 4–7, $n = 0.60$, with $r^2 = 0.9998$; for $Ca^{2+}$ from non-calcined HAP (days 3–7), $n = 0.47$ ($r^2 = 0.9970$); for $Mg^{2+}$ from non-calcined HAPc-5%Sr (days 3–7), $n = 0.52$ ($r^2 = 0.9974$) and from the calcined HAPc-5%Sr sample, $n = 0.61$ ($r^2 = 0.9992$); for phosphate, P, release from calcined HAP (days 3–7), $n = 0.483$, with $r^2 = 0.9951$.

We can infer that in the first immersion days, dissolution processes are predominant ($n$-values are closer to 1), probably assigned to the ions on the surface of NPs, while later on the main process is the inner diffusion of ions from the bulk of the NPs to the interface with the immersion medium. However, the ion exchange process cannot be ruled out.

**Table 4.** Korsmeyer–Peppas model: the $k$ and $n$ constants for cumulated ion release from HAP and ms-HAPs in water and in SBF, in simulated dynamic conditions (for 7 days).

| element | sample | immersion fluid | $10^3\ k$ (day$^{-n}$) | $n$ | $r^2$ |
|---|---|---|---|---|---|
| Ca | HAP non-calcined | water | 1.77 ± 0.12 | 0.644 ± 0.049 | 0.9666 |
| | HAP calcined | | 1.35 ± 0.14 | 0.896 ± 0.076 | 0.9588 |
| | HAPc-5%Sr non-calcined | | 2.14 ± 0.15 | 0.858 ± 0.051 | 0.9794 |
| | HAPc-5%Sr calcined | | 3.45 ± 0.10 | 0.693 ± 0.021 | 0.9947 |
| | HAPc-10%Sr non-calcined | | 2.17 ± 0.14 | 0.863 ± 0.047 | 0.9826 |
| | HAPc-10%Sr calcined | | 2.54 ± 0.11 | 0.685 ± 0.022 | 0.9938 |
| P | HAP non-calcined | water | 6.76 ± 0.49 | 0.857 ± 0.053 | 0.9777 |
| | HAP calcined | | 13.1 ± 0.9 | 0.679 ± 0.049 | 0.9694 |
| | HAPc-5%Sr non-calcined | | 12.9 ± 1.1 | 0.760 ± 0.060 | 0.9638 |
| | HAPc-5%Sr calcined | | 16.4 ± 1.0 | 0.690 ± 0.044 | 0.9760 |
| | HAPc-10%Sr non-calcined | | 9.74 ± 0.78 | 0.836 ± 0.059 | 0.9713 |
| | HAPc-10%Sr calcined | | 10.1 ± 1.0 | 0.858 ± 0.070 | 0.9613 |
| Mg | HAPc-5%Sr non-calcined | water | 39.2 ± 4.7 | 0.808 ± 0.88 | 0.9332 |
| | HAPc-5%Sr calcined | | 33.1 ± 2.4 | 0.775 ± 0.052 | 0.9732 |
| | HAPc-10%Sr non-calcined | | 31.1 ± 1.8 | 0.816 ± 0.041 | 0.9849 |
| | HAPc-10%Sr calcined | | 31.3 ± 1.8 | 0.866 ± 0.041 | 0.9865 |
| Sr | HAPc-5%Sr non-calcined | water | 6.12 ± 0.34 | 0.893 ± 0.040 | 0.9883 |
| | HAPc-5%Sr calcined | | 8.63 ± 0.24 | 0.717 ± 0.020 | 0.9956 |
| | HAPc-10%Sr non-calcined | | 4.40 ± 0.16 | 0.841 ± 0.028 | 0.9995 |
| | HAPc-10%Sr calcined | | 6.46 ± 0.31 | 0.808 ± 0.035 | 0.9891 |
| Sr | HAPc-5%Sr non-calcined | SBF | 12.6 ± 0.6 | 0.831 ± 0.038 | 0.9878 |
| | HAPc-5%Sr calcined | | 15.1 ± 1.0 | 0.826 ± 0.047 | 0.9812 |
| | HAPc-10%Sr non-calcined | | 8.04 ± 0.35 | 0.888 ± 0.032 | 0.9922 |
| | HAPc-10%Sr calcined | | 7.45 ± 0.29 | 0.886 ± 0.028 | 0.9939 |

## 3.6. Potential physiological and medical effects of ion release from multi-substituted hydroxyapatites

The equilibrium of $Ca^{2+}$ and phosphate release/uptake is highly dependent on the simultaneous cation and anion substitution in HAP structure. The gradual increase of the released Ca (figures 4$a$ and 5$b$) and P amount (figures 4$g$ and 5$j$), in water as immersion media, was promoted by the substituted cations and anions in the HAP structure. This is a wanted feature of ms-HAP nanomaterials for bone tissue regeneration, particularly because low solubility and in consequence low resorption rates are described as drawbacks of pure stoichiometric HAP.

The ion release profiles for Mg and Sr are dependent on the cationic and anionic substitution degree in HAP structure, and on the static (figure 4) or simulated dynamic conditions (figure 5). Consequently, different positive effects can appear on the biocompatibility and on the performance of these substituted HAPs, arising from the simultaneous release of bioactive ions from them into the surrounding medium.

In this respect, our previous results showed that the performance of mono-substituted HAP, namely the performance of HAP-10%Sr scaffolds was greater than for HAP-5%Sr scaffolds, which were greater than that for HAP scaffolds in osteoblast culture [21]. This effect was expressed as progressively increases in cell density (cell adhesion) on scaffolds made of these HAP-Sr nanomaterials with the gradually high degree of substitution within HAP lattice. We suggest that the environmental pH and the total concentration of released ions in culture medium assured conditions compatible with cell viability and proliferation.

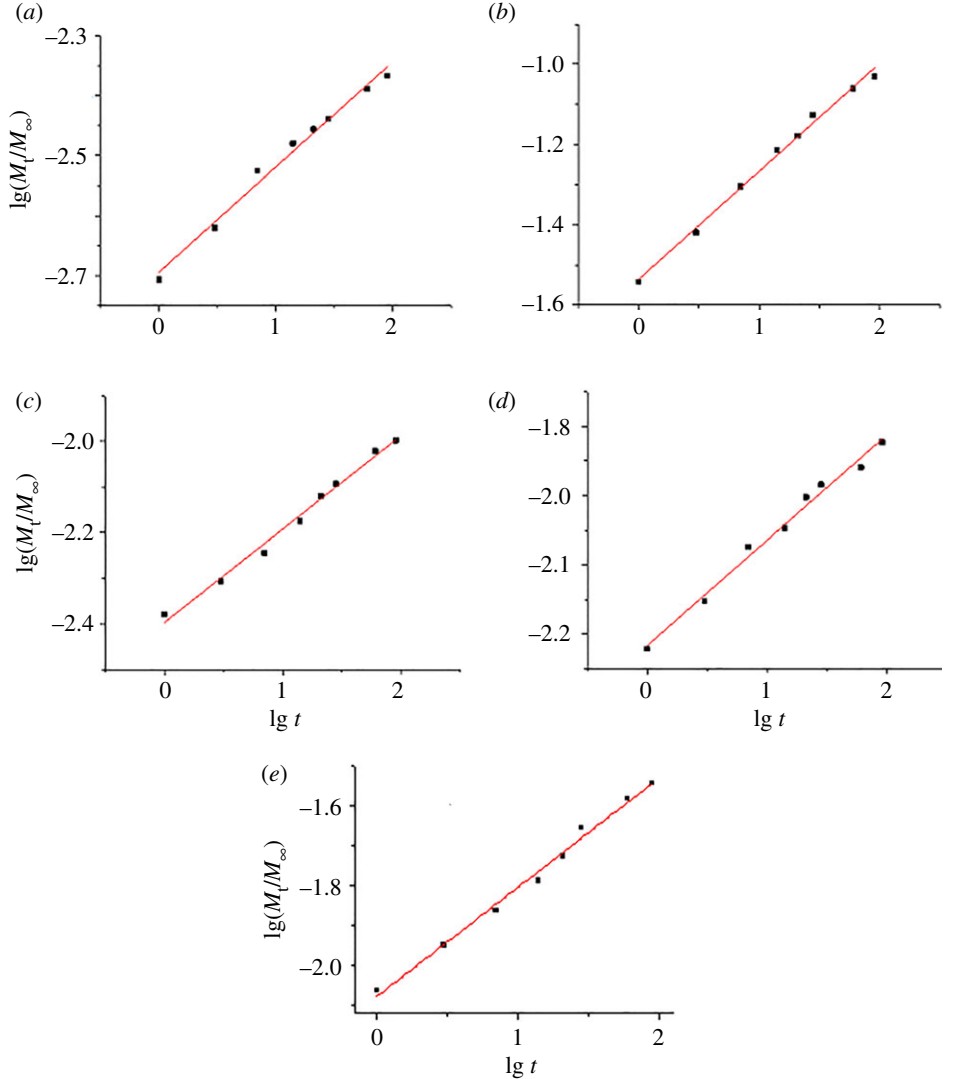

**Figure 7.** Regression lines for ion release of $Ca^{2+}$ (a), $Mg^{2+}$ (b), $Sr^{2+}$ (c) and of phosphate, P, (e), all in water, and the release of $Sr^{2+}$ in SBF (d), from non-calcined HAPc-10%Sr in static conditions, for 90 days; lg stands for the decimal logarithm used in the calculations.

Another example of improved performance of HAP-Mg was attributed to the release of high Mg concentration and its role on integrin-mediated cell adhesion [84]. Zinc oxide-doped HAP, HAP-Zn, biocomposites has shown enhanced mineralization of osteoblasts [88]. The beneficial effects of HAP-Si nanomaterials on osteoblast and osteoclast response were recorded and associated with improved biocompatibility and with enhanced cell activity [14,89].

Our earlier studies showed that HAP-0.6%Mg–0.2%Zn–0.2%Si used as scaffolds in osteoblast cell culture [13] promoted a high cell activity. Also, HAP-1.5%Mg–0.2%Zn–0.2%Si (denoted as HAPc) used as coatings on Ti implants *in vivo* investigation [6] demonstrated an enhanced osseointegration and promoted a fast time-dependent fracture healing. These effects are associated with the time-dependent release of substituting elements within HAP structure.

Moreover, the pH regulation and the control of released amounts of biologically active elements, such as Ca, Mg, Sr, Zn, P and Si, existing in the lattice of these nanomaterials, are important for the most favourable cell culture conditions and are generally requested to promote optimum cell activity. The features of HAPc-5%Sr and HAPc-10%Sr nanomaterials (calcined or non-calcined) are essential for cell adhesion, growth and new bone formation and are in substantial agreement with our preliminary results obtained in osteoblast culture. These discoveries recommend the calcined and non-calcined HAPc-5%Sr and HAPc-10%Sr as potential nanomaterials for therapeutic applications as bone substitutes for bone regeneration and as coatings on orthopaedic and dental metallic implants.

Understanding the impact of multi-substitution in HAP structure and the effect of chemical composition in the structure of ms-HAP materials, and on the ion release from these nanomaterials

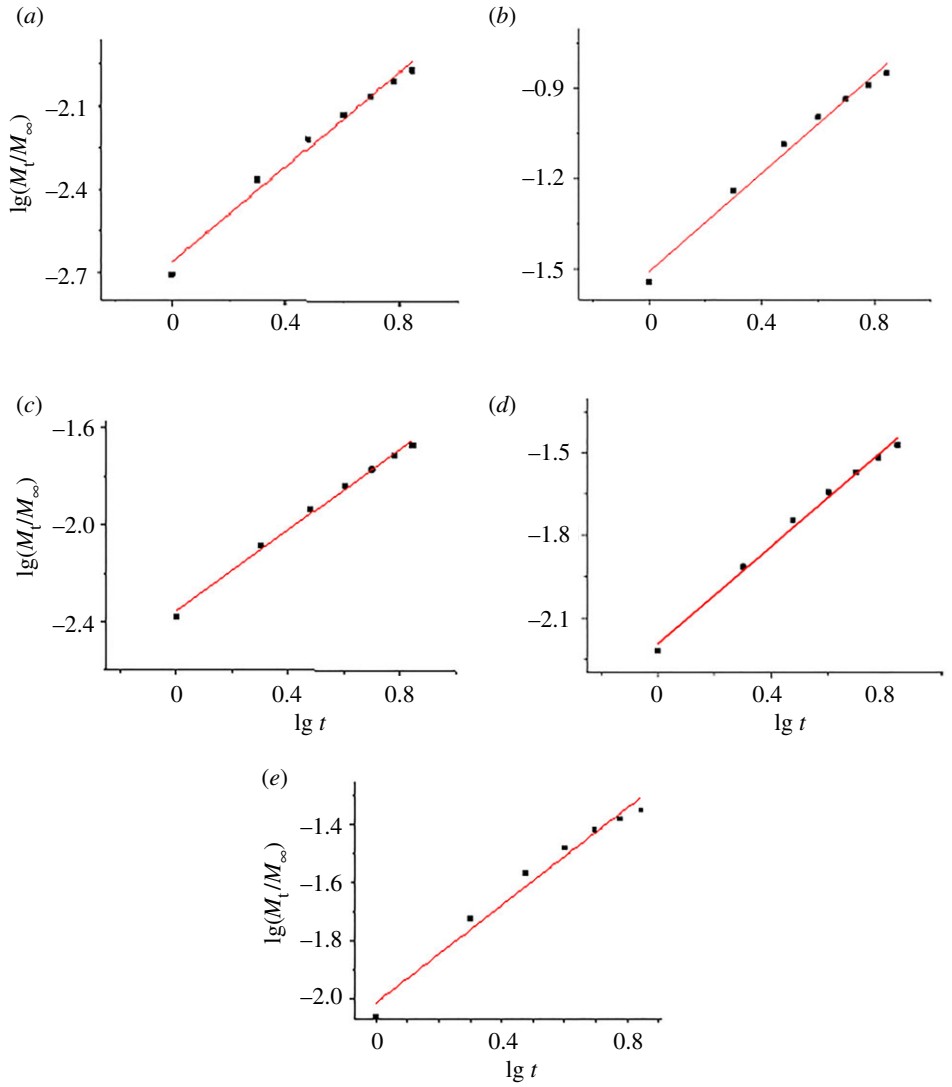

**Figure 8.** Regression lines for release of $Ca^{2+}$ (a), $Mg^{2+}$ (b), $Sr^{2+}$ (c) and of phosphate (e), all in water, and release of $Sr^{2+}$ in SBF (d), from non-calcined HAPc-10%Sr in simulated dynamic conditions, for 7 days; lg stands for the decimal logarithm, used in all calculations.

into surrounding media, brings important knowledge in the advanced development and improvement of designed multifunctional materials for bone regeneration and bone replacement.

Furthermore, the substitution elements will eventually be released into the body, which makes ms-HAP a multifunctional system loaded with multiple biologically active elements [6,13,14,21,24,25]. Considering this complexity, it is to be understood that future studies will be also focused on changes in the biological properties of HAP owing to multiple ion substitution. Otherwise, future research needs to be focused on biological characteristics and function of ms-HAPs, which will be substantiated in *in vivo* experiments.

# 4. Conclusion

The investigated HAP nanomaterials, specifically HAP, HAP-1.5%Mg–0.2%Zn–0.2%Si–5%Sr (HAPc-5% Sr) and HAP-1.5%Mg–0.2%Zn–0.2%Si–10%Sr (HAPc-10%Sr), were synthesized to have a controlled size of NPs, both in non-calcined and calcined powders, to better mimic the natural HAP found in bone structure.

The ion release was determined by the ICP-OES method from these six nanomaterials in water and SBF, in static and simulated dynamic conditions. The released amount was established to be a function of initial concentration of ions both in HAPs and in the surrounding medium. The ion release kinetics was

analysed using the Korsmeyer–Peppas model and it was revealed that the ion release was mainly a diffusion process. This also confirms that ms-HAP biomaterials containing various substituting elements, like Mg, Zn, Sr and Si, can release these active ions progressively in the environment of their graft.

Moreover, the sustained ion release makes them appropriate nanomaterials as bone substitutes for biological and medical applications in orthopaedic and dental areas. Moreover, they would assure a prolonged supply of essential ions of biological importance, indispensable for osteoblast activity and thus can contribute to the formation and development of healthy new bone tissue and bone regeneration. Future research might focus on the consequence of multi-substitution on the structure of ms-HAP nanomaterials and on the ion release from these nanomaterials into surrounding media, to further develop tailored multifunctional nanomaterials for therapeutic applications

Finally, it is rational to anticipate that the HAPc-5%Sr and HAPc-10%Sr nanomaterials, either non-calcined (green powders) or calcined powders, are suitable for bone regeneration and enhanced fracture healing, as already demonstrated [6] with multifunctional HAPc biomimetic coating for *in vivo* experiments.

Data accessibility. Our data are available from the Dryad Digital Repository, https://doi.org/10.5061/dryad.1g1jwstsk [90].
Authors' contributions. A.M. participated in the design of the study, in the characterization of the materials and critically revised the manuscript. O.C. carried out the ICP-OES measurements of concentration for the released ions. P.T.F. helped draft the manuscript and critically revised the manuscript. I.P. carried out the AFM investigations and critically revised the manuscript. G.T. participated in the design of the study and critically revised the manuscript. G.-A.P. contributed in the characterization of the materials (SEM, BET) and helped draft the manuscript. C.P.R. collected data and critically revised the manuscript. O.H. participated in the design of the study, participated in data analysis and the theoretical modelling, helped draft the manuscript and critically revised the manuscript. M.T.-C. conceived of the study, designed the study, coordinated the study, helped draft the manuscript and critically revised the manuscript. All authors gave final approval for publication and agree to be held accountable for the work performed therein.
Competing interests. we declare we have no competing interests.
Funding. This work was financially supported by grants from UEFISCDI, specifically NanoSilva grant no. 43 and Ima-Health PED grant no. 3373.

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
