## [Reviewer comments · Royal Society Open Science]

Review History

RSOS-201119.R0 (Original submission)

Review form: Reviewer 1

Is the manuscript scientifically sound in its present form?

Yes

Are the interpretations and conclusions justified by the results?

Yes

Is the language acceptable?

Yes

Do you have any ethical concerns with this paper?

No

Have you any concerns about statistical analyses in this paper?

No

Recommendation?

Accept as is

Comments to the Author(s)

In my humble opinion, this well-written manuscript may be published as is. No corrections are necessary.

Review form: Reviewer 2

Is the manuscript scientifically sound in its present form?

No

Are the interpretations and conclusions justified by the results?

Yes

Is the language acceptable?

Yes

Do you have any ethical concerns with this paper?

No

Have you any concerns about statistical analyses in this paper?

No

Recommendation?

Accept with minor revision (please list in comments)

Comments to the Author(s)

Comments attached (Appendix A).

Review form: Reviewer 3

Is the manuscript scientifically sound in its present form?

Yes

Are the interpretations and conclusions justified by the results?

No

Is the language acceptable?

Yes

Do you have any ethical concerns with this paper?

No

Have you any concerns about statistical analyses in this paper?

No

Recommendation?

Reject

Comments to the Author(s)

In this manuscript the authors have presented their extensive work on the release profile of bioactive cations and anions from calcined and un-calcined HAp in water and simulated body fluid over prolonged time period. In my opinion such work needs to be substantiated with atleast a few basic in vivo experiments to actually understand its correlation to real time applications. As has been observed by the authors in the current study, the release behavior is largely affected by the concentration of the cation/anion incorporated in the base material as well as the ionic concentration of the fluid in which the pellets are suspended. Hence, the significance of such studies can be substantiated atleast some basic animal studies. Also, studies on biocompatibility of the material by MTT etc also need to be carried out.

In the introduction, the authors state that embedded ions contribute towards bioactivity in terms of bone regeneration and antimicrobial activity. Do the authors expect to see both these activities in similar range of concentration of incorporated ions?

Decision letter (RSOS-201119.R0)

Dear Dr Tomoaia-Cotisel:

Manuscript ID: RSOS-201119

Title: "Ions release from hydroxyapatite and substituted hydroxyapatites in different immersion liquids. In vitro experiments and theoretical modelling study"

Thank you for submitting the above manuscript to Royal Society Open Science. Your paper was sent to reviewers and their comments are included at the bottom of this letter.

In view of the concerns raised by the reviewers, the manuscript has been rejected in its current form. However, a new manuscript may be submitted which takes into consideration these comments.

Please note that resubmitting your manuscript does not guarantee eventual acceptance, and that your resubmission will be subject to peer review before a decision is made.

Your resubmitted manuscript should be submitted by 10-Feb-2021. If you are unable to submit by this date please contact the Editorial Office.

On behalf of the Subject Editor Professor Anthony Stace and the Associate Editor Mr Andrew Dunn

REVIEWER(S) REPORTS:

Associate Editor Comments to Author ():

RSC Associate Editor:

Comments to the Author:

(There are no comments.)

RSC Subject Editor:

Comments to the Author:

(There are no comments.)

Reviewers' Comments to Author:

Reviewer: 1

Comments to the Author(s)

In my humble opinion, this well-written manuscript may be published as is. No corrections are necessary.

Reviewer: 2

Comments to the Author(s)

Comments attached

Reviewer: 3

Comments to the Author(s)

In this manuscript the authors have presented their extensive work on the release profile of bioactive cations and anions from calcined and un-calcined HAp in water and simulated body fluid over prolonged time period. In my opinion such work needs to be substantiated with at least a few basic in vivo experiments to actually understand its correlation to real time applications. As has been observed by the authors in the current study, the release behavior is largely affected by the concentration of the cation/anion incorporated in the base material as well as the ionic concentration of the fluid in which the pellets are suspended. Hence, the significance of such studies can be substantiated at least some basic animal studies. Also, studies on biocompatibility of the material by MTT etc also need to be carried out.

In the introduction, the authors state that embedded ions contribute towards bioactivity in terms of bone regeneration and antimicrobial activity. Do the authors expect to see both these activities in similar range of concentration of incorporated ions?

Author's Response to Decision Letter for (RSOS-201119.R0)

See Appendix B.

RSOS-201785.R0

Review form: Reviewer 2

Is the manuscript scientifically sound in its present form?

Yes

Are the interpretations and conclusions justified by the results?

Yes

Is the language acceptable?

Yes

Do you have any ethical concerns with this paper?

No

Have you any concerns about statistical analyses in this paper?

No

Recommendation?

Accept as is

Comments to the Author(s)

Recommended for publication

Decision letter (RSOS-201785.R0)

Dear Dr Tomoaia-Cotisel:

Title: Ions release from hydroxyapatite and substituted hydroxyapatites in different immersion liquids. In vitro experiments and theoretical modelling study

Manuscript ID: RSOS-201785

It is a pleasure to accept your manuscript in its current form for publication in Royal Society Open Science. The chemistry content of Royal Society Open Science is published in collaboration with the Royal Society of Chemistry.

RSC Associate Editor
Comments to the Author:
(There are no comments.)

Reviewer(s)' Comments to Author:
Reviewer: 2

Comments to the Author(s)
Recommended for publication

Appendix A

Reviewer Comments

Manuscript Title : Ions release from hydroxyapatite and substituted hydroxyapatites in different immersion liquids. *In vitro* experiments and theoretical modeling study

Manuscript ID : RSOS-201119

In this manuscript, hydroxyapatite (HAP) and multiple ions substituted hydroxyapatites (HAPc-5%Sr and HAPc-10%Sr) were prepared. The hydroxyapatite containing 1.5 wt% of Mg, 0.2 wt% of Zn and 0.2 wt% of Si is denoted as HAPc. The prepared materials were calcined at 300 °C for 1 h. The samples obtained before (noncalcined: (1) HAP, (2) HAPc-5%Sr and (3) HAPc-10%Sr) and after (calcined: (4) HAP, (5) HAPc-5%Sr and (6) HAPc-10%Sr) calcination were characterized by SEM, AFM and BET measurements. The *in vitro* release of cations and anions of both calcined and noncalcined samples derived from nanosized HAP and substituted HAPs (HAPc-5%Sr and HAPc-10%Sr) were evaluated in water and SBF solution under static and simulated dynamic conditions by using ICP-OES technique. However, some questions and suggestions are recommended to improve the manuscript.

1. In Experimental section, add the detail information of the mole concentration and amount of each chemical used for the sample preparation.
2. It is better to add the reason behind the specific (wt%) selection of Mg, Zn, and Si (1.5 wt% Mg, 0.2 wt% Zn, 0.2 wt% Si).
1. Please mentioned that how to calculate the theoretical formulas for HAPc-5%Sr as $\text{Ca}_{8.76}\text{Mg}_{0.63}\text{Zn}_{0.03}\text{Sr}_{0.58}(\text{PO}_4)_{5.93}(\text{SiO}_4)_{0.07}(\text{OH})_{1.93}$ and HAPc-10%Sr as $\text{Ca}_{8.12}\text{Mg}_{0.65}\text{Zn}_{0.03}\text{Sr}_{1.20}(\text{PO}_4)_{5.93}(\text{SiO}_4)_{0.07}(\text{OH})_{1.93}$.
2. Sample names should be clearly indicated in the Experimental section for each study.
3. Write the equation “ $\lg(\text{Mt} / \text{M}\alpha) = \lg k + n \lg t$ ” clearly as “ $\log (\text{Mt} / \text{M}\alpha) = \log k + n \log t$ ”.
4. Please add the corresponding EDS spectrum and EDS elemental mapping for the SEM analysis to differentiate the elements present in HAP and multiple ions substituted HAP (viz., Mg, Zn, Si, Sr, P, etc.).

5. SEM analysis part needs more and clear discussion about the morphologies variation before and after calcination.
6. Roughness value should be added as mentioned in page:8.
7. The morphologies of the noncalcined samples and the corresponding nanoparticles size looks good than those obtained after calcination. Specify the reason.

Appendix B

Manuscript Title : Ions release from hydroxyapatite and substituted hydroxyapatites in different immersion liquids. *In vitro* experiments and theoretical modeling study

Manuscript ID : RSOS-201119

Reviewer 1:

Comments to the Author(s)

In my humble opinion, this well-written manuscript may be published as is. No corrections are necessary.

We respectfully thank to Reviewer 1 for the positive evaluation of our manuscript.

Reviewer 2

Comments

Manuscript Title: Ions release from hydroxyapatite and substituted hydroxyapatites in different immersion liquids. *In vitro* experiments and theoretical modeling study

Manuscript ID: RSOS-201119

In this manuscript, hydroxyapatite (HAP) and multiple ions substituted hydroxyapatites (HAPc-5%Sr and HAPc-10%Sr) were prepared. The hydroxyapatite containing 1.5 wt% of Mg, 0.2 wt% of Zn and 0.2 wt% of Si is denoted as HAPc. The prepared materials were calcined at 300 °C for 1 h. The samples obtained before (noncalcined: (1) HAP, (2) HAPc-5%Sr and (3) HAPc-10%Sr) and after (calcined: (4) HAP, (5) HAPc-5%Sr and (6) HAPc-10%Sr) calcination were characterized by SEM, AFM and BET measurements. The *in vitro* release of cations and anions of both calcined and noncalcined samples derived from nanosized HAP and substituted HAPs (HAPc-5%Sr and HAPc-10%Sr) were evaluated in water and SBF solution under static and simulated dynamic conditions by using ICP-OES technique. However, some questions and suggestions are recommended to improve the manuscript.

1. In Experimental section, add the detail information of the mole concentration and amount of each chemical used for the sample preparation.

In agreement with the reviewer's recommendation, we added the detailed data about the mole concentration and amount of each chemical used for the sample preparation, in Experimental section of revised manuscript (highlighted in yellow).

1. Please mentioned that how to calculate the theoretical formulas for HAPc-5%Sr as $\text{Ca}_{8.76}\text{Mg}_{0.63}\text{Zn}_{0.03}\text{Sr}_{0.58}(\text{PO}_4)_{5.93}(\text{SiO}_4)_{0.07}(\text{OH})_{1.93}$ and HAPc-10%Sr as $\text{Ca}_{8.12}\text{Mg}_{0.65}\text{Zn}_{0.03}\text{Sr}_{1.20}(\text{PO}_4)_{5.93}(\text{SiO}_4)_{0.07}(\text{OH})_{1.93}$.

We appreciated the reviewer suggestion and we have shown the way to calculate the theoretical formulas (as a new section in Materials and Methods - highlighted in yellow) for HAPc-5%Sr and HAPc-10%Sr. This calculation is important in terms of data comparison in the state of the art, but it is usually missing in the state of the art, even for mono substituted HAP and co-substituted HAP. We hope that this calculation will stimulate researchers to apply it in their research and development of the new generation of multiple substituted hydroxyapatites, as biomimetic ceramics, for bone regeneration and bone healing.

This research will continue to be of growing interest, particularly as researchers develop innovative methods to fabricate multifunctional materials for bone regeneration

2. Sample names should be clearly indicated in the Experimental section for each study.

Each sample name is clearly indicated in the Experimental section in agreement with reviewer's suggestion; **highlighted in yellow**.

2. It is better to add the reason behind the specific (wt%) selection of Mg, Zn, and Si (1.5 wt% Mg, 0.2 wt% Zn, 0.2 wt% Si).

We thank the reviewer for the suggestion and the opportunity for us to add the reason for the selection of the complex hydroxyapatite, noted HAPc, as HAP-1.5 wt% Mg- 0.2 wt% Zn - 0.2 wt% Si, and consequently to describe the rational design of HAPc-5 wt% Sr and HAPc-10 wt% Sr. This aspect is given in the Introduction section of revised R1 manuscript, **highlighted in yellow**.

3. Write the equation " $\lg(M_t / M_\alpha) = \lg k + n \lg t$ " clearly as " $\log (M_t / M_\alpha) = \log k + n \log t$ ".

In agreement with the reviewer's recommendation, the equation was written as requested by the reviewer (**highlighted in yellow**).

$$\log (M_t / M_\alpha) = \log k + n \log t$$

According to ISO 80 000 - 1: 2009,. $\lg k$ is the recommended notation for the decimal logarithm (common logarithm), which was used in our calculations, as noted in the captions of Figs.7 and 8.

4. Please add the corresponding EDS spectrum and EDS elemental mapping for the SEM analysis to differentiate the elements present in HAP and multiple ions substituted HAP (viz., Mg, Zn, Si, Sr, P, etc.).

We are thankful for the reviewer's comment. Multi-colored SEM image, showing the multiple elements in the multi-substituted HAP, for HAPc-5%Sr calcined nano powders, and the corresponding EDS spectrum, jointly with mono-colored SEM images, showing every specific element to distinguish their homogeneous distribution within the ms-HAP, were included in Fig. 1G, H, I, as requested. The interpretation of these data is also included in the R1 manuscript (**highlighted in yellow**).

5. SEM analysis part needs more and clear discussion about the morphologies variation before and after calcination.

In agreement with reviewer suggestion a discussion about the morphologies variation before and after calcination was added at Fig. 1. (**highlighted in yellow**).

6. Roughness value should be added as mentioned in page:8.

Roughness values are given in the legend of Fig. 2, and a remark is given on page 8. (**highlighted in yellow**).

7. The morphologies of the noncalcined samples and the corresponding nanoparticles size looks good than those obtained after calcination. Specify the reason.

In agreement with reviewer suggestion also given in comment no. 5 - a discussion about the morphologies variation before and after calcination was added jointly with the explanation behind this aspect is also given. (highlighted in yellow)

Reviewer 3

In this manuscript the authors have presented their extensive work on the release profile of bioactive cations and anions from calcined and un-calcined HAP in water and simulated body fluid over prolonged time period.

- 1) “In my opinion such work needs to be substantiated with at least a few basic in vivo experiments to actually understand its correlation to real time application.”

We thank the reviewer 3 for the opinion.

Respectfully, the objective of the present work is on ions release from ms-HAPs – as said in the title –and biological effects in vitro and in vivo will be studied in the future. Thus, we added in this R- manuscript a paragraph before **Conclusion** section, showing the importance of this future investigation.

We emphasize that our presented previous results in this R-manuscript describe successfully the **potential physiological and medical effects of ions release from ms-HAPs**.

In the following we show some data from our previous research in vitro using cell culture and in vivo using rats' model.

As known and recognized, this type of work is very time consuming and is very expensive working with animals, as well as a big research team is involved, namely chemists, physicists, material scientists, engineers, surgeons, clinical physicians, biophysicists, molecular biologists, medical doctors, etc.

Fortunately, we recently published about the enhancement of bone consolidation using titanium implants coated with biomimetic composite, which contains HAP_c= HAP-1.5%Mg-0.2%Zn-0.2%Si *in vivo* evaluation using rat model.⁶ Ti implants coated by biomimetic composite comprising HAP_c enhanced the bone consolidation process, thus potentially providing a superior strategy for clinical applications.

“These results suggest that in vivo experiments essentially/actually the ions release from biomimetic coatings, comprising HAP-1.5%Mg-0.2%Zn-0,2%Si, is enough to promote and maintain bone regeneration as well as sustains and demonstrates an understanding of its correlation to real time application. This discovery is in total agreement with the opinion of the reviewer 3: “In my opinion such work needs to be substantiated with at least a few basic in vivo experiments to actually understand its correlation to real time application”.

Certainly, this HAP_c coating has the ability to bond with the bone tissue and so improves the implant stability and may reduce the healing time after surgery. The *in vivo* uses of HAP_c-5%Sr or HAP_c-10%Sr will have at least similar effects with HAP_c or eventually the elevated effects particularly on osteoporosis.

Respectfully, the importance of ions release from ms-HAPs is given in the section “Potential physiological and medical effects of ions release from ms-HAPs”.

2) As has been observed by the authors in the current study, the release behavior is largely affected by the concentration of the cation/anion incorporated in the base material as well as the ionic concentration of the fluid in which the pellets are suspended.

- a) Hence, the significance of such studies can be substantiated at least some basic animal studies.

The effects of HAPc were excellent against Ti implants uncovered by biomimetic nanomaterials. Results demonstrate that this HAPc coating has the ability to bond with the bone tissue and so improves the implant stability and may reduce the healing time after surgery.⁶ Please see the answer given above.

This discovery⁶ supports the significance of such studies substantiated in the animal studies, as reviewer 3 suggested.

- b) Also, studies on **biocompatibility** of the material by MTT etc also need to be carried out.

Biocompatibility of HAPc was already evaluated by ion release in vivo study, cell attachment and proliferation of osteoblasts as well as differentiation of osteoblasts to osteocytes during the formation of new bone in the healing process of fractures. Obviously, HAPc-5%Sr and HAPc-10%Sr will have a similar effect as HAPc.

Alternatively, the total effect of all substituting elements might be greater than the sum of the individual effects of each substituting element in the HAP lattice. A synergistic effect can be beneficial for new bone formation particularly in the treatment of osteoporosis. The bioactivity and biocompatibility of HAP-Sr scaffolds was proven by us in osteoblasts culture, by MTT and increased expression of alkaline phosphatase activity including also osteocalcin.²¹

Also, a related nanomaterial with HAPc, namely HAP-0.6%Mg-0.2%Zn-0.2%Si, even at low concentration in Mg confirmed a high bioactivity in osteoblasts culture by MTT, and collagen production.¹³

The last paragraph in Conclusion section was completed with previous results on HAPc in vivo study.⁶

Finally, it is rational to anticipate that the HAPc-5%Sr and HAPc-10%Sr nanomaterials, either noncalcined (green powders) or calcined powders, are suitable for bone regeneration and enhanced fracture healing, as already demonstrated⁶ with multifunctional HAPc biomimetic coating in vivo experiments. (highlighted in yellow).

3) In the introduction, the authors state that embedded ions contribute towards bioactivity in terms of bone regeneration and antimicrobial activity. Do the authors expect to see both these activities in similar range of concentration of incorporated ions?

Our results demonstrate that embedded ions contribute towards bioactivity in terms of bone regeneration in similar range of concentration of incorporated ions, as shown in our previous published work, both *in vivo* studies and *in vitro* studies on similar substituted HAPs. Regarding antimicrobial activity, we used HAPs as carriers of AgNPs with remarkable results against different pathogens.²⁹

The following paragraph was added in the paper – about the future research - just before the **Conclusion** section (highlighted in yellow):

Furthermore, the substitution elements will eventually be released into the body, which makes multi-substituted hydroxyapatite as a multifunctional system loaded with multiple biologically active elements.^{6,13,14,21,24,25} Considering this complexity, it is to be understood that the future studies will be also focused on changes in the biological properties of HAP due to the multiple ions substitution. Otherwise, the future research needs to be focused on biological characteristics and function of ms-HAPs, which will be substantiated in vivo experiments.